# An Analysis for Reasoning Bias of Language Models with Small Initialization

**Junjie Yao** [1]   **Zhongwang Zhang** [1]   **Zhi-Qin John Xu** [2 3 4]

## Abstract

Transformer-based Large Language Models (LLMs) have revolutionized Natural Language Processing by demonstrating exceptional performance across diverse tasks. This study investigates the impact of the parameter initialization scale on the training behavior and task preferences of LLMs. We discover that smaller initialization scales encourage models to favor reasoning tasks, whereas larger initialization scales lead to a preference for memorization tasks. We validate this reasoning bias via real datasets and meticulously designed anchor functions. Further analysis of initial training dynamics suggests that specific model components, particularly the embedding space and self-attention mechanisms, play pivotal roles in shaping these learning biases. We provide a theoretical framework from the perspective of model training dynamics to explain these phenomena. Additionally, experiments on real-world language tasks corroborate our theoretical insights. This work enhances our understanding of how initialization strategies influence LLM performance on reasoning tasks and offers valuable guidelines for training models.

## 1. Introduction

With the rapid advancement of deep learning technologies, Large Language Models have achieved remarkable success in the field of Natural Language Processing (NLP). These models have demonstrated exceptional capabilities across a

[1]School of Mathematical Sciences, Institute of Natural Sciences, Shanghai Jiao Tong University, Shanghai, P.R. China [2]Institute of Natural Sciences, School of Mathematical Sciences, MOE-LSC, School of Artificial Intelligence, Shanghai Jiao Tong University, Shanghai, P.R. China [3]Center for LLM, Institute for Advanced Algorithms Research, Shanghai, P.R. China [4]Shanghai Seres Information Technology Co., Ltd, Shanghai 200040, China. Correspondence to: Zhongwang Zhang <0123zzw666@sjtu.edu.cn>, Zhi-Qin John Xu <xuzhiqin@sjtu.edu.cn>.

*Proceedings of the 42$^{nd}$ International Conference on Machine Learning*, Vancouver, Canada. PMLR 267, 2025. Copyright 2025 by the author(s).

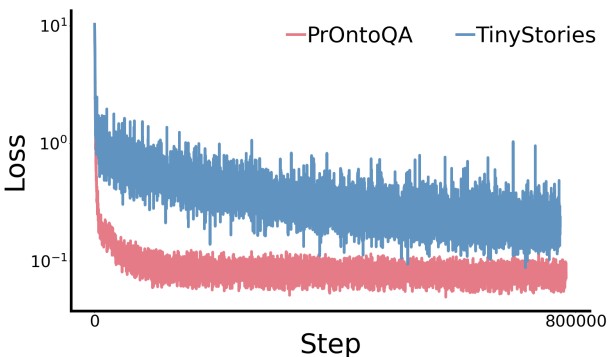

*Figure 1.* Comparison of training loss between PrOntoQA and TinyStories in one next-token prediction training for this mix dataset. The red line represents the training loss on the PrOntoQA dataset, while the blue line depicts the training loss on the TinyStories dataset.

wide range of tasks, from text generation to complex reasoning (Wei et al., 2022a; Achiam et al., 2023; Liu et al., 2024). Reasoning, in particular, is a critical ability for LLMs. A number of studies have focused on improving the reasoning ability of these models through data-driven approaches, such as RHO-1 (Lin et al., 2024) and Phi-4 (Abdin et al., 2024). However, there remains an ongoing debate as to whether LLMs genuinely learn the underlying logical rules or merely mimic patterns observed in the data (Marcus, 2003; Smolensky et al., 2022).

An alternative approach to enhancing the reasoning ability of LLMs focuses on the model architecture and its training process. In one such study examining the use of Transformers to model compositional functions, it was observed that the scale of model parameter initialization significantly impacts the model's reasoning behavior (Zhang et al., 2024a; 2025). Specifically, smaller initialization scales bias the model toward fitting the data by learning primitive-level functions and compositional rules, whereas larger initialization scales tend to encourage memorization of input-output mappings. A qualitative rationale for this phenomenon has been proposed: with a small initialization, a well-documented effect known as neuron condensation emerges during training (Xu et al., 2025; Luo et al., 2021; Zhou et al., 2022; Zhang et al., 2022; Zhang & Xu, 2023; Zhang et al., 2023; Zhang & Xu, 2024). This phenomenon suggests that neurons within the same layer tend to behave

similarly, promoting data fitting with the least possible complexity. To achieve a low-complexity result, the model must learn a minimal set of rules leading to capture the intrinsic primitive functions and compositional rules. However, this rationale does not reveal a critical question: how the optimization process, together with the Transformer structure, can achieve reasoning solutions with small initialization?

In this work, we identify a reasoning bias during the training of neural networks that learn natural language when initialized with small parameter scales. To illustrate this phenomenon, we employ a GPT-2 model (Radford et al., 2019) to train on a mixed dataset comprising two types of language data with distinct levels of reasoning complexity, within a single next-token prediction training framework. The first dataset, PrOntoQA (Saparov & He, 2023), consists of question-answering examples that include chains of thought, which explicitly describe the reasoning necessary to answer the questions correctly. The second dataset, TinyStories (Eldan & Li, 2023), is a synthetic corpus of short stories containing only words typically understood by children aged 3 to 4 years. As shown in Figure 1, the training loss for PrOntoQA decreases significantly faster than for TinyStories, suggesting that the model encounters and learns the reasoning patterns more readily.

We uncover a key mechanism whereby reasoning tasks are learned earlier during training because the tokens associated with these tasks become more differentiated in the embedding space at an early stage of the training process. We validate this mechanism using both synthetic data and real-world datasets. Furthermore, we provide a theoretical explanation for the evolution of token embeddings, which depends on the distribution of sample labels. Since each token is encoded as a one-hot vector, its embedding is adjusted based on the loss associated with the labels of that token. Consequently, different label distributions can lead to distinct learning behaviors for token embeddings. For memory tasks, the labels associated with each token are typically random and lack explicit structure, which results in similar distributions for different memory token labels. As a result, the embeddings for memory tokens are difficult to differentiate in the early stages of training. In contrast, reasoning tokens often exhibit distinct label distributions, leading to more differentiated embedding vectors for these tokens. These insights are elaborated through a simplified model using a multi-layer perceptron (MLP) and embedding structure, followed by an analysis of a Transformer model.

The primary contribution of this research lies in uncovering the impact of the parameter initialization scale on the training behavior and task preferences of LLMs. By combining theoretical analysis with empirical evidence, we enhance the understanding of LLM training dynamics and provide new insights for optimizing model initialization strategies.

## 2. Preliminary

### 2.1. Synthetic Composition Task

To study the task bias during the training, we use the concept of anchor function (Zhang et al., 2024b) to construct a dataset that contains tasks of different reasoning complexities. We consider all tokens belonging to positive integers. A set of tokens are designated as anchors, denoted as $\mathcal{A} := \{a \in \mathbb{N}^+ | \alpha_{\min} \leq a \leq \alpha_{\max}\}$, where each anchor represents an addition/randomness operation in this work. Another set of tokens are designated as keys, denoted as $\mathcal{Z} := \{z \in \mathbb{N}^+ | \zeta_{\min} \leq z \leq \zeta_{\max}\}$ with the assumption that $\mathcal{Z} \cap \mathcal{A} = \emptyset$. For convenience, we denote $N_{\mathcal{Z}} = \zeta_{\max} - \zeta_{\min} + 1$ and $N_{\mathcal{A}} = \alpha_{\max} - \alpha_{\min} + 1$.

This section introduces two types of sequence mappings. The first step involves constructing a sequence of positive integers with length $L$, represented as:

$$\mathcal{X}^{(q,L)} = \{X | X = [z_1, \cdots, z_p, a_{p+1}, \\ \cdots, a_{p+q}, z_{p+q+1}, \cdots, z_L], z_i \in \mathcal{Z}, a_i \in \mathcal{A}\}. \quad (1)$$

We define $q$ as the number of anchors in the sequence, and $p$ as the index of the element immediately preceding the first anchor element $a_{p+1}$ in the sequence.

For a given sequence $X \in \mathcal{X}^{(q,L)}$, we define its key-anchor combination as $(z_p, a_{p+1}, \cdots, a_{p+q})$, which is denoted concisely as pair $(z_p, \boldsymbol{a})$, and other keys are regarded as noise in this input sequence. The anchor set $\mathcal{A}$ is divided into two subsets, i.e., reasoning anchor set $\mathcal{A}_{\mathrm{rsn}}$ and memory anchor set $\mathcal{A}_{\mathrm{mem}}$, where $\mathcal{A} = \mathcal{A}_{\mathrm{rsn}} \cup \mathcal{A}_{\mathrm{mem}}$ and $\mathcal{A}_{\mathrm{rsn}} \cap \mathcal{A}_{\mathrm{mem}} = \emptyset$.

**Reasoning mapping.** For any $X$ with $a_{p+i} \in \mathcal{A}_{\mathrm{rsn}}, i = 1, \cdots, q$, we define the following mapping as a reasoning mapping

$$\mathcal{F}_{\mathrm{rsn}}(X) = z_p + \sum_{i=1}^{q} a_{p+i}.$$

**Memory mapping.** For any key-anchor pair $(z_p, \boldsymbol{a})$, where each element in $\boldsymbol{a}$ belongs to $\mathcal{A}_{\mathrm{mem}}$, we randomly sample a number $y^{(z_p, \boldsymbol{a})}$ from $\mathcal{Z}$ as the memory mapping label of any sequence $X$ containing $(z_p, \boldsymbol{a})$, i.e.

$$\mathcal{F}_{\mathrm{mem}}(X) = y^{(z_p, \boldsymbol{a})}, \quad \forall X \text{ contains } (z_p, \boldsymbol{a}).$$

A detailed example is provided in Figure 2. It's noted that the key-anchor pair may occur at any position within the sequence. The label is independent of both the noise tokens and the position of the key-anchor pair within the sequence but is determined solely by the value of the key-anchor pair.

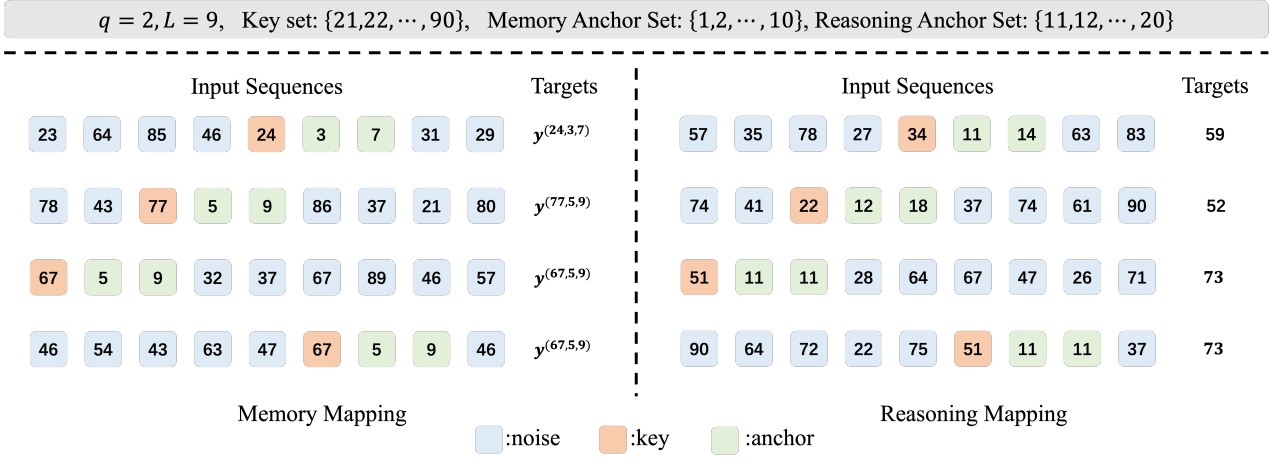

*Figure 2.* Schematic diagram of the synthetic composition task. The gray-shaded area illustrates the specific setup used in this example. Each block represents a token within the input sequence, with different face colors indicating distinct token types (blue: noise, orange: key, green: anchor). Each row corresponds to an input sequence paired with its respective label. The left section depicts four examples of memory mapping, while the right section presents four examples of reasoning mapping.

## 2.2. Dataset Setup

In this study, we denote a data pair as $(X, y)$, where $X$ represents the input sequence and $y$ corresponds to its associated label. We define $\boldsymbol{y}$ as the one-hot encoded representation of $y$ for convenience. For memory mapping, all data are contained within the training set $\mathcal{D}_{\mathrm{mem}}$, and no test set is employed, as the generalization is not considered in this framework. For reasoning mapping, we define a set of masked anchor combinations $\mathcal{M} = \{(a_{p+1}, a_{p+2}, \cdots, a_{p+q}) \mid a_{p+i} \in \mathcal{A}_{\mathrm{rsn}}, i = 1, \cdots, q\}$ and designate all sequences containing any masked combination $(a_{p+1}, a_{p+2}, \cdots, a_{p+q}) \in \mathcal{M}$ as the test set $\mathcal{D}_{\mathrm{rsn,test}}$ and set the rest sequence as $\mathcal{D}_{\mathrm{rsn,train}}$. The training set is $\mathcal{D}_{\mathrm{train}} = \mathcal{D}_{\mathrm{mem}} \cup \mathcal{D}_{\mathrm{rsn,train}}$.

## 2.3. Model Architecture

We give the formulation of the embedding space and self-attention module here for notation convenience. Let $d_{\mathrm{vob}}, d_m, d_k$ denote the vocabulary size, embedding space dimension, and query-key-value projection dimension, respectively. For any token $s$, denote its one-hot vector by $\boldsymbol{e}^s \in \mathbb{R}^{1 \times d_{\mathrm{vob}}}$. The embedding vector of $s$ is $\boldsymbol{w}^{\mathrm{emb},s} = \boldsymbol{e}^s \boldsymbol{W}^{\mathrm{emb}}$ where $\boldsymbol{W}^{\mathrm{emb}} \in \mathbb{R}^{d_{\mathrm{vob}} \times d_m}$ is the embedding matrix. Additionally, the self-attention operator $\mathrm{Attn}$ on any embedding sequence $\boldsymbol{X} \in \mathbb{R}^{L \times d_m}$ is defined as:

$$\mathrm{Attn}\left(\boldsymbol{X}\right) = g\left(\mathrm{mask}\left(\frac{\boldsymbol{X}\boldsymbol{W}^Q\boldsymbol{W}^{KT}\boldsymbol{X}^T}{\sqrt{d_k}}\right)\right), \quad (2)$$

$$\boldsymbol{O} = \mathrm{Attn}\left(\boldsymbol{X}\right)\boldsymbol{X}\boldsymbol{W}^V\boldsymbol{W}^O, \quad (3)$$

where $g\left(\cdot\right)$ is the softmax operator and $T$ means the matrix transposition. $\boldsymbol{W}^Q, \boldsymbol{W}^K, \boldsymbol{W}^V \in \mathbb{R}^{d_m \times d_k}$ are the query, key and value projection matrices, respectively. $\boldsymbol{W}^O \in$

$\mathbb{R}^{d_k \times d_m}$ represents the output projection matrix. The detailed expression of multilayer Transformer models can be found in Appendix A.1.

## 2.4. Parameter Initialization

Given any trainable parameter matrix $\boldsymbol{W} \in \mathbb{R}^{d_1 \times d_2}$, where $d_1$ and $d_2$ denote the input and output dimensions, respectively, its elements are initialized according to a normal distribution:

$$\boldsymbol{W}_{i,j} \sim \mathcal{N}\left(0, \left(d_1^{-\gamma}\right)^2\right),$$

where $\gamma$ is the initialization rate. Specifically, the initialization scale decreases as $\gamma$ increases. Note that $\gamma = 0.5$ is commonly used in many default initialization methods, such as LeCun initialization (LeCun et al., 1998) and He initialization (He et al., 2015). As the network width towards infinity (Luo et al., 2021; Zhou et al., 2022), the training of the network with $\gamma > 0.5$ exhibits significant non-linear characteristics, i.e., condensation. Therefore, initialization scales with $\gamma > 0.5$ are generally considered small.

## 3. Result

In this section, we present empirical evidence of a reasoning bias during the training of Transformers with small initialization by utilizing composition tasks. To further explore this phenomenon, we introduce a simplified model consisting of an embedding layer and a multi-layer perceptron, which reproduces the reasoning bias and enables theoretical analysis. A key mechanism underlying this bias is that the training behavior of each token's embedding depends on the label distribution of the samples containing that token. For reasoning anchors, the label distributions typically exhibit

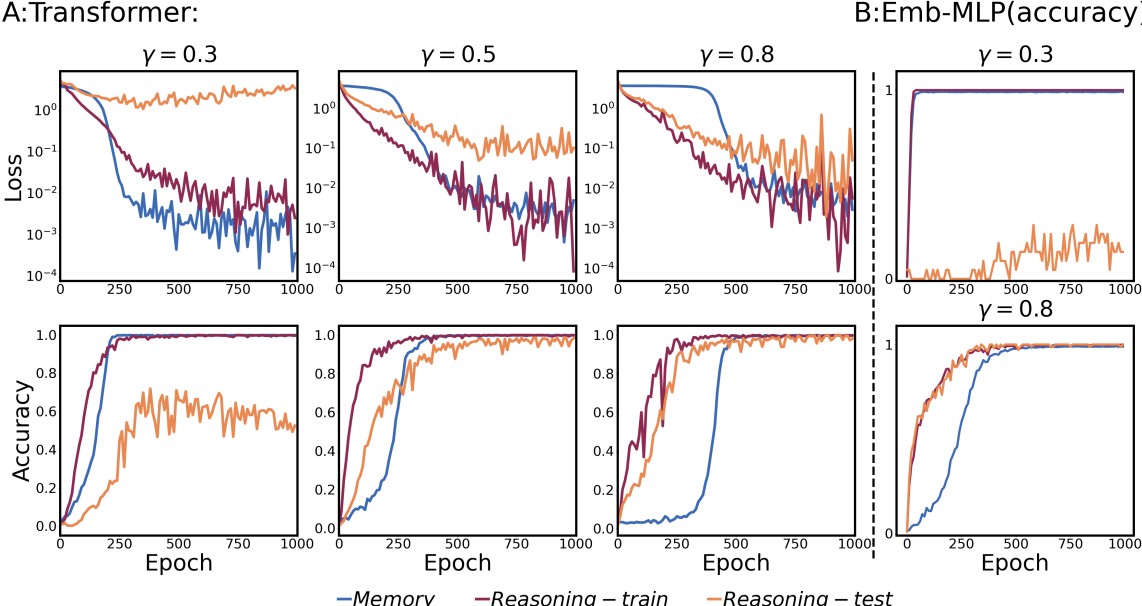

*Figure 3.* A: Loss and prediction accuracy of the models on different datasets under varying initialization scales ($\gamma = 0.3, 0.5, 0.8$). The top row depicts the evolution of the loss during training for three datasets: $\mathcal{D}_{\mathrm{mem}}$ (blue lines), $\mathcal{D}_{\mathrm{rsn,train}}$ (purple lines), and $\mathcal{D}_{\mathrm{rsn,test}}$ (orange lines). The bottom row presents the corresponding prediction accuracies for these datasets. Each column represents results obtained with different initialization scales. B: Prediction accuracy of Emb-MLP under initialization rate $\gamma = 0.3$ and $\gamma = 0.8$.

greater variability compared to memory anchors, leading to the more rapid differentiation of their embeddings early in training. Additionally, we extend our analysis to the Transformer architecture to demonstrate the generalizability of this effect. For each observation mentioned in the following, we provide a similar analysis with larger initialization scales in Appendix C.

### 3.1. Reasoning Bias in Transformer with Composite Anchor Functions

In our experiment, we set that $q = 2, L = 9$. The dataset is constructed with the following configurations: $\mathcal{Z} = \{21, \cdots, 120\}, \mathcal{A}_{\mathrm{mem}} = \{1, \cdots, 10\}, \mathcal{A}_{\mathrm{rsn}} = \{11, \cdots, 20\}$ and $\mathcal{M} = \{(11, 13), (13, 11)\}$. The dataset contains 200000 data pairs, ensuring an equal number of samples for each anchor combination. The loss function employed is Cross Entropy and the optimization algorithm used is AdamW. The model architecture comprises a decoder-only Transformer structure with 2 layers and a single attention head. We train the model under different initialization scales with $\gamma = 0.3, 0.5, 0.8$ utilizing the last token prediction method. Additional details about the experimental setup can be found in Appendix A.

To investigate the impact on training behavior under varying initialization scales, we analyze the dynamics of loss and prediction accuracy on $\mathcal{D}_{\mathrm{mem}}, \mathcal{D}_{\mathrm{rsn,train}}$ and $\mathcal{D}_{\mathrm{rsn,test}}$. As illustrated in Figure 3A, for $\gamma = 0.3$, the losses on $\mathcal{D}_{\mathrm{mem}}$ and $\mathcal{D}_{\mathrm{rsn,train}}$ decrease at nearly identical rates, while the

loss on $\mathcal{D}_{\mathrm{rsn,test}}$ remains effectively unchanged. This observation suggests that the model primarily memorizes the training data in this setting. In contrast, when $\gamma = 0.8$, the losses on $\mathcal{D}_{\mathrm{rsn,train}}$ and $\mathcal{D}_{\mathrm{rsn,test}}$ decrease significantly faster than the loss on $\mathcal{D}_{\mathrm{mem}}$. This behavior indicates a shift towards a reasoning bias in the model. These findings reveal that the model's learning bias is influenced by the initialization scale: as the initialization scale decreases, the model exhibits a progressively stronger reasoning bias.

### 3.2. Simplified Model: Phenomena and Analysis

To further investigate the underlying cause of the reasoning bias under a small initialization scale, we begin by employing a two-layer fully connected network to address a particular task, where $p \equiv 1$ and $L = q + 1$. The network structure is defined as follows:

**Definition 1.** *Given that* $\boldsymbol{W}^{(1)} \in \mathbb{R}^{d_m \times d_f}, \boldsymbol{W}^{(2)} \in \mathbb{R}^{d_f \times d_{\mathrm{vob}}},$ *and* $\sigma$ *as the activation function. Given any sequence* $X \in \mathcal{X}^{(q,q+1)}$, *we define the Embedding-MLP model (Emb-MLP)* $\boldsymbol{G}_{\boldsymbol{\theta}}$ *as*

$$\boldsymbol{G}_{\boldsymbol{\theta}}(X) := \sigma\left(\sum_{s \in X} \boldsymbol{w}^{\mathrm{emb},s} \boldsymbol{W}^{(1)}\right) \boldsymbol{W}^{(2)}.$$

Comparing with a large initialization scale ($\gamma = 0.3$), a noticeable reasoning bias can still be observed in Figure 3B for a small initialization scale ($\gamma = 0.8$).

**Embedding space exhibits distinct patterns** To investigate the causes of the reasoning bias under small initialization for such a simplified model, it's critical to understand the structure of the embedding space. Figure 4A depicts the cosine similarity matrices for embeddings of memory anchors and reasoning anchors at epochs 50 and 900. The results reveal that the cosine similarity between reasoning anchors $s_i, s_j$ decreases with the increase of $|s_i - s_j|$, suggesting that reasoning anchors quickly establish a hierarchical structure within the embedding space. In contrast, the memory anchors exhibit consistently high similarity and alignment, leading to a lack of differentiation among them. Nevertheless, given that the model needs to learn more primitive-level mappings for memory mapping than reasoning mapping, the embedding space associated with memory mapping should, in principle, exhibit greater complexity and variability. This phenomenon reveals that the primary challenge preventing the model from effectively learning memory mapping could highly possibly lie in its difficulty in identifying and differentiating between individual anchors.

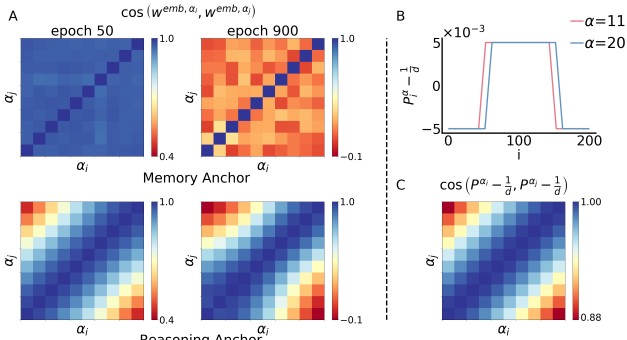

*Figure 4.* A: Cosine similarity matrices for memory (top row) and reasoning (bottom row) anchors at epoch 50 (left) and epoch 900 (right) of a model initialized with $\gamma = 0.8$. B: Distribution of $\boldsymbol{P}^s - \frac{1}{d_{\text{vob}}}\mathbf{1}$ for different reasoning anchor $s$. C: Cosine similarity between $\boldsymbol{P}^{s_i} - \frac{1}{d_{\text{vob}}}\mathbf{1}$ and $\boldsymbol{P}^{s_j} - \frac{1}{d_{\text{vob}}}\mathbf{1}$ for any reasoning anchor $s_i, s_j$, exhibiting a similar structure to the embedding space of reasoning anchors observed in A.

**Target distribution shapes the embedding** This phenomenon can be interpreted through the training dynamics. To facilitate our analysis, we give the following assumption (Chen et al., 2024):

**Assumption 1.** *The activation function $\sigma \in \mathcal{C}^2(\mathbb{R})$, and there exists a universal constant $C_L > 0$ such that its first and second derivatives satisfy $\|\sigma'(\cdot)\|_\infty \leq C_L, \|\sigma''(\cdot)\|_\infty \leq C_L$. Moreover, $\sigma(0) = 0, \sigma'(0) = 1$.*

For any token $s$, let $\left\{\left(X^{s,i}, y^{s,i}\right)\right\}_{i=1}^{n_s}$ denote all input sequences containing $s$ and corresponding labels, where $n_s$ means the appearance times of $s$. As the initialization scale decreases, with Assumption 1, we have

$\sigma'\left(\sum_{x \in X^i} \boldsymbol{w}^{\text{emb},x}\boldsymbol{W}^{(1)}\right) = \mathbf{1}, \text{softmax}\left(\boldsymbol{G_\theta}\left(X^{s,i}\right)\right) = \frac{1}{d_{\text{vob}}}\mathbf{1}$, where $\mathbf{1} \in \mathbb{R}^{1 \times d_{\text{vob}}}$ means the vector with all elements equal to 1. Then the gradient flow of $\boldsymbol{w}^{\text{emb},s}$ could be approximated by the limit formulation, i.e.,

$$\frac{d\boldsymbol{w}^{\text{emb},s}}{dt} = \frac{1}{n}\sum_{i=1}^{n_s}\left(\boldsymbol{y}^{s,i} - \frac{1}{d_{\text{vob}}}\mathbf{1}\right)\boldsymbol{W}^{(2)T}\boldsymbol{W}^{(1)T}, \quad (4)$$

where $n$ represents the count of all tokens. We consider $n \to \infty$ to obtain the asymptotic form of the following gradient flow.

**Proposition 1.** *For any token $s$, denote $Y^s$ as a random variable, which takes values randomly from the label of any input sequence that contains token $s$. In the limit $n \to \infty$, we define $\boldsymbol{P}^s$ with its $i$-th element as the probability of $Y^s = i$, i.e., $\boldsymbol{P}_i^s = \mathbb{P}(Y^s = i)$. Assume the ratio of the token $s$ in the whole dataset $r_s := \frac{n_s}{n}$ remains constant, then (4) can be approximated as:*

$$\frac{d\boldsymbol{w}^{\text{emb},s}}{dt} = r_s\left(\boldsymbol{P}^s - \frac{1}{d_{\text{vob}}}\mathbf{1}\right)\boldsymbol{W}^{(2)T}\boldsymbol{W}^{(1)T}. \quad (5)$$

The proof is provided in Appendix B.1. Proposition 1 demonstrates that for any token $s$, its embedding vector is dominated by the distribution of $Y^s$ which indicates that it's significant to discuss the distribution $Y^s$ for different tokens. Firstly, we define the following random variables ($\mathcal{U}$ for discrete uniform distribution):

$$Z \sim \mathcal{U}(\mathcal{Z}), \quad A_j \sim \mathcal{U}(\mathcal{A}_{\text{rsn}}), \quad j = 1, 2, \cdots, q. \quad (6)$$

Then we have the following results:

$$\mathbb{P}(Y^s = i \mid s \in \mathcal{A}_{\text{mem}}) = \frac{1}{N_{\mathcal{Z}}}\delta_{i \in \mathcal{Z}}, \quad (7)$$

and

$$\mathbb{P}(Y^s = i \mid s \in \mathcal{A}_{\text{rsn}})$$
$$=\mathbb{P}\left(Z + \sum_{j=1}^{q-1} A_j = i - s \mid s \in \mathcal{A}_{\text{rsn}}\right). \quad (8)$$

Equation (7) reveals that the information to different memory anchors is identical such that the embedding space of memory anchors exhibits a high similarity. However, (8) demonstrates that the distribution of $Y^s$ exhibits shifts in the mean values that depend on the specific $s$ for any $s \in \mathcal{A}_{\text{rsn}}$. Figure 4B and 4C visualize the distribution of $\boldsymbol{P}^s - \frac{1}{d_{\text{vob}}}\mathbf{1}$ and the resulting cosine similarity among different reasoning anchors $s$, suggesting that the labels' distributions play a critical role in establishing the embedding structure of reasoning anchors during the early stages of training, facilitating the differentiation among the embeddings associated with different reasoning anchors. The detailed formulations can be found in Appendix B.2.

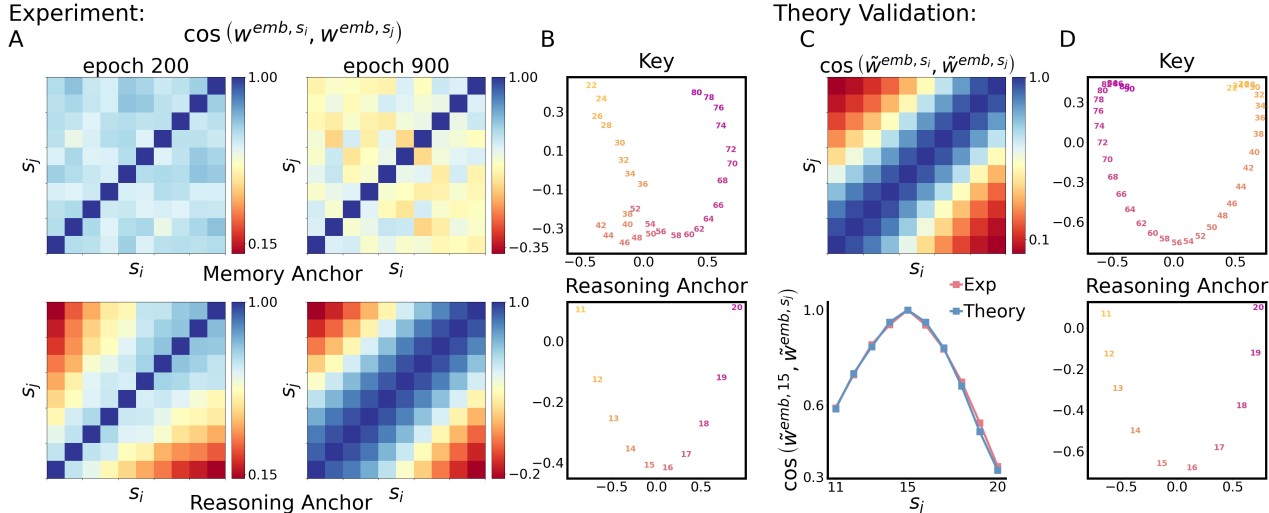

*Figure 5.* Embedding structure of a Transformer model with small initialization scale. A: Cosine similarity matrices for memory (top) and reasoning (bottom) anchors at epoch 200 (left) and epoch 900 (right). B: Visualization of the embedding space projected onto the first two principal components computed via PCA. C: Cosine similarity between the constructed embedding vectors of reasoning anchors $\tilde{\boldsymbol{w}}^{\mathrm{emb,s}}$ (see (12)) as derived in Theorem 1 (top) and Cosine similarity comparison between experimental results with theoretical approximations where $s_i = 15$ (bottom). D: PCA projection of the constructed embedding space $\tilde{\boldsymbol{w}}^{\mathrm{emb,s}}$ for $s \in \mathcal{Z}$ (top) and $s \in \mathcal{A}_{\mathrm{rsn}}$ (bottom) onto the first two principal components.

### 3.3. Transformer with General Task

In the previous section, we investigate the key mechanisms driving the learning bias of Emb-MLP and analyze the dynamics of its embedding space. However, when applied to a general sequence containing some degree of noise, i.e., $L > q + 1$, we find the MLP model fails to perform effectively due to its inability to extract critical tokens $z_p, a_{p+1}$, and $a_{p+2}$. In contrast, Transformer architectures overcome this limitation through self-attention mechanisms, which can identify the key and anchor elements and propagate their information.

In the following section, we conduct an in-depth analysis of the Transformer's characteristics and processing mechanisms under a small initialization scale. Specifically, we investigate whether the embedding space exhibits similar phenomena to those observed in Emb-MLP and assess how the model captures critical information from the input sequence.

**Embedding space.** The embedding space of the Transformer exhibits a phenomenon similar to that observed in the Emb-MLP. Figure 5A illustrates the cosine similarity among different anchors' embedding vectors, revealing distinct patterns for reasoning and memory tasks. Reasoning anchors display a hierarchical structure, the further distance, the smaller similarity, suggesting a clearer organization within the embedding space. In contrast, memory anchors exhibit high similarity and alignment. Additionally, we apply Principal Component Analysis (PCA) to the entire embedding

space to examine its structural properties. The results in Figure 5B reveal a strong inherent numerical order. This structure is particularly advantageous for reasoning tasks, as it supports the model's capacity to generalize based on the underlying numerical relationships.

**First attention module.** As illustrated in Figure 6, the first attention matrix approaches the behavior of an average operator when initialized with small scales, such that $(\mathrm{Attn}\,(\boldsymbol{X})\,\boldsymbol{V})_i = \frac{1}{i} \sum_{j \leq i} \boldsymbol{V}_j$, where $\boldsymbol{V} = \boldsymbol{X}\boldsymbol{W}^V$. Consequently, each token aggregates information from all preceding tokens. Additionally, the largest singular value of $\boldsymbol{W}^V$ is significantly larger than the remaining singular values, and its corresponding singular vector is aligned closely with the embedding vectors of reasoning anchors, but nearly orthogonal to those of memory anchors. These phenomena suggest reasoning anchors are prominently captured by $\boldsymbol{W}^V$ and subsequently propagated to all subsequent tokens in the sequence via the average operation. However, the memory anchors are not distinctly identified, indicating that the model faces challenges in capturing significant information from a memory sequence effectively. More analysis of $\boldsymbol{W}^V$ can be found in Appendix B.8.

**Second attention module.** The second attention module functions to extract the key, and propagate its information to the final position in the sequence. This is facilitated through the use of position embeddings. Since this mechanism is applicable to both memory and reasoning tasks, we provide a detailed explanation in Appendix D.

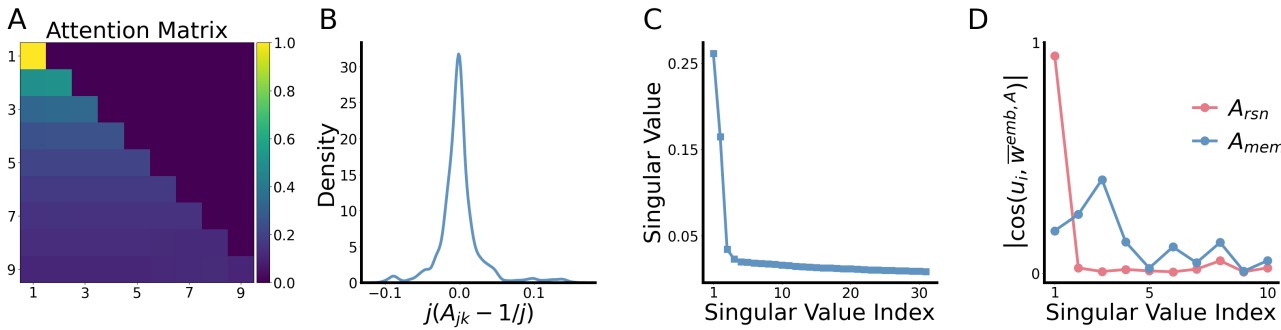

*Figure 6.* Characteristics of the first attention module under small initialization ($\gamma = 0.8$) in the early training stage (epoch 200). A: Heatmap of the attention matrix for a random sample. B: Distribution of the relative error between attention $A_{jk}$ and $\frac{1}{j}$ across all training sequences. C: Distribution of singular values of $\boldsymbol{W}^V$. D: Cosine similarity between the left singular vectors and average embedding vectors of the anchors.

**Theoretical analysis.** Based on the observations from the experiments, we extract the sketch component of the model, which is crucial to its learning preferences, and analyze the underlying mechanisms for its occurrence. We define the following one-layer Transformer model:

**Definition 2** (One-layer Transformer). *Let $d_f \in \mathbb{N}^+$ denotes the hidden layer of the feedforward neural network (FNN). For any $X \in \mathcal{X}^{(q,L)}$, denote $\mathrm{Attn}\left(\boldsymbol{W}^{\mathrm{emb},X}\right)$ by $\boldsymbol{A}$, then we define $\boldsymbol{f_\theta} : \mathcal{X}^{(q,L)} \to \mathbb{R}^{d_m}$ as follows:*

$$\boldsymbol{f_\theta}(X) = \sigma((\boldsymbol{A}_{L,:}\boldsymbol{W}^{\mathrm{emb},X}\boldsymbol{W}^V\boldsymbol{W}^O + \boldsymbol{W}^{\mathrm{emb},X}_{L,:})\boldsymbol{W}^{f1})\boldsymbol{W}^{f2}$$
$$+ \boldsymbol{A}_{L,:}\boldsymbol{W}^{\mathrm{emb},X}\boldsymbol{W}^V\boldsymbol{W}^O + \boldsymbol{W}^{\mathrm{emb},X}_{L,:},$$
$$(9)$$

*where $\boldsymbol{W}^{f1} \in \mathbb{R}^{d_m \times d_f}, \boldsymbol{W}^{f2} \in \mathbb{R}^{d_f \times d_m}$ are the feedforward layer projection matrices. The subscript $L,:$ in $\boldsymbol{A}_{L,:}$ and $\boldsymbol{W}^{\mathrm{emb},X}_{L,:}$ denotes the $L$-th row.*

Definition 2 is introduced to facilitate the theoretical analysis, excluding the Layer Normalization and the final projection operator, as they do not impact our results (see Appendix F).

As we observed, with a small initialization scale, the attention operator $\boldsymbol{A}$ can be interpreted as an average operator. Specifically, we have

**Lemma 1.** *For any $\varepsilon \in (0, 1]$, there exists $C > 0$ such that for any $\gamma > C$, the elements of $\boldsymbol{A}$ at initialization, denoted by $\boldsymbol{A}_{i,j}$, satisfy $\left|\boldsymbol{A}_{i,j} - \frac{1}{i}\right| \leq \varepsilon$ for any $i \leq j$ with probability at least $1 - \varepsilon$.*

Denote that $\boldsymbol{W}^f = \boldsymbol{W}^{f1}\boldsymbol{W}^{f2}, \boldsymbol{W}^{VO} = \boldsymbol{W}^V\boldsymbol{W}^O$ and $\tilde{\boldsymbol{W}} = \left(\boldsymbol{W}^{f,T} + \boldsymbol{I}\right)\left(\boldsymbol{W}^{VO,T} + \boldsymbol{I}\right)$, where the identity matrix $\boldsymbol{I}$ comes from the resnet. Using techniques similar to those employed in the previous section, we derive the gradient flow of $\boldsymbol{w}^{\mathrm{emb},s}$ under small initialization scales as follows:

**Proposition 2.** *For any $s \in \mathcal{A}_{\mathrm{mem}}$, let $n, \gamma \to \infty$, with*

*Assumption 1 we have the following result:*

$$\frac{d\boldsymbol{w}^{\mathrm{emb},s}}{dt} = \frac{r_s}{L}\left(\frac{\boldsymbol{\delta}^{\mathcal{Z}}}{N_{\mathcal{Z}}} - \frac{1}{d_m}\boldsymbol{1}\right)\tilde{\boldsymbol{W}}.$$
$$(10)$$

**Proposition 3.** *For any $s \in \mathcal{A}_{\mathrm{rsn}}$, let $n, \gamma \to \infty$, with Assumption 1 we have the following result:*

$$\frac{d\boldsymbol{w}^{\mathrm{emb},s}}{dt} = \frac{r_s}{L}\left(\boldsymbol{P}^s - \frac{1}{d_m}\boldsymbol{1}\right)\tilde{\boldsymbol{W}},$$
$$(11)$$

*where the $i$-th element of $\boldsymbol{P}^s$ is defined as $\boldsymbol{P}^s_i = \mathbb{P}\left(Z + \sum_{j=1}^{q-1} A_j = i - s \mid s \in \mathcal{A}_{\mathrm{rsn}}\right)$.*

Furthermore, we utilize the normal distribution to approximate the distribution of $Y^s, s \in \mathcal{A}_{\mathrm{rsn}}$ and give an approximation of $\boldsymbol{w}^{\mathrm{emb},s}$ to describe the overall structure and internal relationships within the embedding space observed in real-world training scenarios.

**Theorem 1.** *let $n \to \infty$, define the learning rate $\eta$ and assume that $L - q = O(1)$, $\frac{r_s\eta}{L} = O(1)$ and $\|\boldsymbol{w}^{\mathrm{emb}}\|_\infty \leq O\left(d_m^{-\gamma}\right)$ at initialization. We propose the approximation of $\boldsymbol{w}^{\mathrm{emb},s}, s \in \mathcal{A}_{\mathrm{rsn}}$ by*

$$\tilde{\boldsymbol{w}}^{\mathrm{emb},s}_j = C_1\left(C_2 e^{-\frac{(j-s)^2}{2\sigma_P}} - \frac{1}{d_m}\right) + \varepsilon,$$
$$(12)$$

*where $C_1, C_2, \sigma_P$ are constants depending on $r_s, \eta, L, q$ and $\varepsilon \sim \mathcal{N}(0, (d_m^{-\gamma})^2)$. Then we have the following result*

$$\sup_{i,j}\left|\left(\tilde{\boldsymbol{w}}^{\mathrm{emb},s_j}, \tilde{\boldsymbol{w}}^{\mathrm{emb},s_i}\right) - \left(\boldsymbol{w}^{\mathrm{emb},s_j}, \boldsymbol{w}^{\mathrm{emb},s_i}\right)\right|$$
$$\leq O\left(d_m^{1-\gamma}\left(q^{-\frac{1}{2}} + d_m^{-\gamma}\right)\right),$$
$$(13)$$

*where $(\cdot, \cdot)$ denotes the inner production.*

Additionally, for any key $z \in \mathcal{Z}$ and $\mathcal{A}_{mem}$, we could have a similar result. To validate our theory analysis, we set the

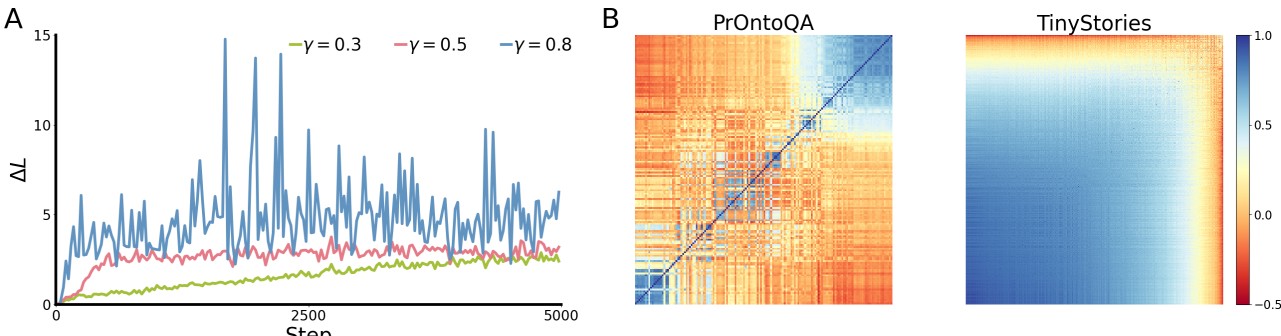

*Figure 7.* Reasoning bias of GPT-2 in real language tasks. A: dynamics of $\Delta L$ during early training stage with initializations scales $\gamma = 0.3, 0.5$ and $0.8$. B: Cosine similarity among embedding space of PrOntoQA dataset and TinyStories dataset at step 5000 with $\gamma = 0.8$.

detailed formulation of reasoning anchors and keys as

$$
\begin{aligned}
\tilde{\boldsymbol{w}}_i^{\mathrm{emb},s} &= e^{-\frac{(i-s)^2}{12}} - \frac{1}{d_m} + \varepsilon, \quad s \in \mathcal{A}_{\mathrm{rsn}}, \\
\tilde{\boldsymbol{w}}_i^{\mathrm{emb},s} &= e^{-\frac{(i-s)^2}{12}} + \varepsilon, \quad s \in \mathcal{Z}.
\end{aligned}
\tag{14}
$$

Figure 5C exhibits the cosine similarity among the $\tilde{\boldsymbol{w}}^{\mathrm{emb},s}$ for any $s \in \mathcal{A}_{\mathrm{rsn}}$ (top) and compare $\cos\left(\boldsymbol{w}^{\mathrm{emb},15}, \boldsymbol{w}^{\mathrm{emb},s_j}\right)$ in real training process with the theoretical approximation $\cos\left(\tilde{\boldsymbol{w}}^{\mathrm{emb},15}, \tilde{\boldsymbol{w}}^{\mathrm{emb},s_j}\right)$ (bottom, a complete comparison is provided in Appendix B.7). Figure 5D presents the PCA projection of $\boldsymbol{w}^{\mathrm{emb},s}$ for $s \in \mathcal{A}_{\mathrm{rsn}}$ and $\mathcal{Z}$, respectively. These visualizations exhibit a strong alignment with the experimental observations, thereby substantiating the validity of our analysis. The proofs of our theoretical results are provided in Appendix B.4, B.5, and B.6.

### 3.4. Real Language Tasks

For the experiment in Figure 1, we also conduct comparisons with initialization scales $\gamma = 0.3$ and $0.5$. To quantify the reasoning bias of the model, we define the following metric:

$$
\Delta L := \frac{L_{\mathrm{Tinystories}} - L_{\mathrm{PrOntoQA}}}{L_{\mathrm{PrOntoQA}}},
$$

where $L_{\mathrm{Tinystories}}$ and $L_{\mathrm{PrOntoQA}}$ denote the loss on TinyStories and PrOntoQA, respectively. As $\gamma$ increases, $\Delta L$ exhibits an upward trend, indicating a growing bias for reasoning task, which is depicted in Figure 7A. To further validate our analysis, we examine the embedding space of the GPT-2 model during the early stages of training which we train with a small initialization scale. Figure 7B reveals that the embeddings of tokens in PrOntoQA are significantly more distinguishable from each other compared to the tokens in TinyStories. The average cosine similarity among the PrOntoQA is 0.123 while 0.531 among the TinyStories. These results provide strong support for our analysis, highlighting the impact of embedding distinguishability on training preference.

### 3.5. Effect of Label's Distribution

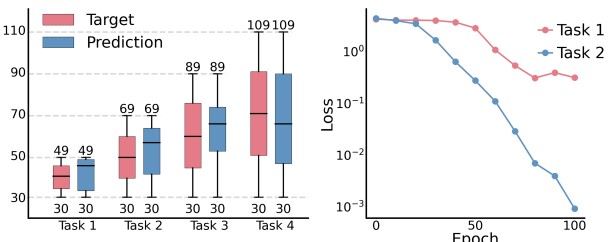

*Figure 8.* Left: Distribution of targets and predictions in 4 groups of memory tasks. Red represents the target distribution in each group and blue represents the prediction distribution. Right: Learning speed comparison for $\mathcal{F}_{\mathrm{mem},1}$ (red) and $\mathcal{F}_{\mathrm{mem},2}$ (blue).

Previous sections reveal that under small initialization, the distribution of labels plays a pivotal role in shaping the embedding space of tokens and regulating the model's training dynamics. To more intuitively demonstrate the impact of the label distribution of each token on its output, we designed four groups of memory mappings. The label ranges of the four groups are set to $\{30, \cdots, 29 + 20 \times i\}, i = 1, 2, 3, 4$. The right picture of Figure 8 illustrates the distribution of the model's outputs for each group during the early stages of training. Notably, it can be observed that even at this initial stage, when the model's accuracy is still relatively low, its outputs do not exceed the range of the label distributions. This highlights the critical influence of label distributions on the token structure, which in turn significantly impacts the model's outputs. Additionally, we compare two memory tasks with differing label distributions. In the first task, denoted $\mathcal{F}_{mem,1}$, for any key-anchor pair $(z_p, \boldsymbol{a})$, the label $y^{(z_p, \boldsymbol{a})}$ is randomly sampled from $\mathcal{Z}$. In the second task $\mathcal{F}_{mem,2}$, $y^{(z_p, \boldsymbol{a})}$ is randomly sampled from $\left\{ z_p - \sum_{i=p+1}^{p+q} a_i, \cdots, z_p + \sum_{i=p+1}^{p+q} a_i \right\}$. While both tasks are clearly memory tasks, the label distributions in $\mathcal{F}_{mem,2}$ vary depending on the anchor. As shown in the left picture of Figure 8, the learning rate for $\mathcal{F}_{mem,2}$

is significantly faster than that for $\mathcal{F}_{mem,1}$. This observation underscores the crucial role that label distribution plays in the model's learning process.

## 4. Related Works

Recent advancements in large language models have shown remarkable capabilities, often surpassing human-level performance in many tasks (Fu et al., 2022; Wei et al., 2022a). However, despite their strong performance in many aspects (Srivastava et al., 2022), LLMs face challenges in handling complex reasoning tasks (Csordás et al., 2021; Dziri et al., 2024; Hupkes et al., 2018; Lepori et al., 2023; Okawa et al., 2023; Yun et al., 2022; Wang et al., 2024d; Csordás et al., 2022). For example, Ramesh et al. (Ramesh et al., 2023) show that Transformers trained on a synthetic benchmark struggle when tasked with combining multiple reasoning steps. Similarly, Liu et al. (Liu et al., 2022) suggest that shallow Transformers tend to learn shortcuts during training, which limits their ability to perform well in more complex reasoning scenarios. Several strategies have been proposed to address these challenges, such as encouraging the generation of explicit reasoning steps in a single output (Wei et al., 2022b) or using LLMs to iteratively produce reasoning steps (Creswell et al., 2022; Creswell & Shanahan, 2022). Despite these efforts, achieving reliability remains a significant challenge. Additionally, some studies have explored the internal mechanisms of language models to enhance their performance (Wang et al., 2024b;c; 2023), but they often do not address the impact of training dynamics on the model's final behavior. To better understand these models' behaviors and inner workings, Zhang et al. (Zhang et al., 2024b) introduced anchor functions as a tool for probing Transformer behavior. Building on this framework, our research investigates how different initialization scales influence model reasoning bias and internal mechanisms from a perspective of training dynamics.

The fitting ability and generalization of deep learning models are critical in understanding deep learning (Breiman, 1995; Zhang et al., 2016), and the initialization of neural network parameters plays a crucial role in determining the network's fitting results (Arora et al., 2019; Chizat & Bach, 2018; Zhang et al., 2019b; E et al., 2020; Jacot et al., 2018; Mei et al., 2018; Rotskoff & Vanden-Eijnden, 2018; Sirignano & Spiliopoulos, 2020; Williams et al., 2019). Luo et al. (Luo et al., 2021) and Zhou et al. (Zhou et al., 2022) primarily identify the linear and condensed regimes in wide ReLU neural networks. In the condensed regime, neuron weights within the same layer tend to become similar. A body of research indicates that condensed networks often exhibit strong generalization capabilities (Zhang et al., 2022; Zhang & Xu, 2023; Zhang et al., 2023; Zhang & Xu, 2024). See (Xu et al., 2025) for an overview of condensation. In our study, we demonstrate that with small initialization values, the parameters of the embedding layer can reach a low-rank state rather than a condensed state. This means that while embeddings of different tokens become linearly dependent, they are not identical. This distinction allows low-rank models to effectively capture essential patterns and generalize well without the stringent alignment required by condensation, which is particularly important for applications such as word embedding matrices where distinct representations for different tokens are necessary. Recent investigations have also explored how initialization affects the training dynamics of LLMs (Huang et al., 2020; Liu et al., 2020; Trockman & Kolter, 2023; Wang et al., 2024a; Zhang et al., 2019a; Zhu et al., 2021). These studies mainly examine how the scale of initialization influences the stability of the training process and is vital for ensuring efficient and effective training of LLMs. In our work, we observe that different initialization schemes result in varying speeds of convergence for memorization tasks versus reasoning tasks and provide a theoretical rationale for this behavior.

## 5. Conclusions

In this paper, we investigate the underlying mechanism of which small initialization scales promote a reasoning preference in language models. Our findings suggest that the label distribution of tokens plays a pivotal role in shaping the embedding space, thereby influencing the learning dynamics and task complexity. Our result can be readily extended to the next-token prediction training and obtain similar results. This perspective is supported through a combination of experimental observations and theoretical analysis, providing a deeper understanding of how initialization strategies impact task-specific behavior in language models.

## Acknowledgements

This work is supported by the National Key R&D Program of China Grant No. 2022YFA1008200, the National Natural Science Foundation of China Grant No. 92270001(Z. X.), 12371511 (Z. X.), 12422119 (Z. X.), Shanghai Municipal of Science and Technology Major Project No. 2021SHZDZX0102 (Z. X.), and the HPC of School of Mathematical Sciences and the Student Innovation Center, and the Siyuan-1 cluster supported by the Center for High Performance Computing at Shanghai Jiao Tong University, and Key Laboratory of Marine Intelligent Equipment and System (Ministry of Education, P.R. China), and SJTU Kunpeng & Ascend Center of Excellence.

## Impact Statement

Our works provide new insights into the intrinsic mechanisms underlying the reasoning bias of language models

under small initialization scales, as well as the training behavior of individual modules within the model architecture. These findings not only contribute to understanding the model behavior and training mechanisms, but also help with optimizing model initialization strategies and designing novel algorithms to enhance the reasoning capabilities of language models.

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

# A. Basic Settings

## A.1. Transformer Architecture

For any sequence $X \in \mathcal{X}^{(q,L)}$, we denote its one-hot vector by $e^X$. The word embedding $\boldsymbol{W}^{\mathrm{emb}}$ and the input to the first Transformer block $X^{(1)}$ is calculated as:

$$\boldsymbol{W}^{\mathrm{emb},X} = e^X \boldsymbol{W}^{\mathrm{emb}}, \quad \boldsymbol{X}^{(1)} = \boldsymbol{W}^{\mathrm{emb},X} + \boldsymbol{W}^{\mathrm{pos}}, \tag{15}$$

where $\boldsymbol{W}^{\mathrm{pos}}$ is a trainable positional vector. For the $l$-th layer, the $\boldsymbol{Q}, \boldsymbol{K}, \boldsymbol{V}$ are defined as:

$$\boldsymbol{Q}^{(l)} = \boldsymbol{X}^{(l)} \boldsymbol{W}^{Q(l)}, \quad \boldsymbol{K}^{(l)} = \boldsymbol{X}^{(l)} \boldsymbol{W}^{K(l)}, \quad \boldsymbol{V}^{(l)} = \boldsymbol{X}^{(l)} \boldsymbol{W}^{V(l)}. \tag{16}$$

The attention matrix $\mathrm{Attn}^{(l)}$ and its subsequent output $\boldsymbol{X}^{\mathrm{qkv}(l)}$ for the $l$-th layer is computed as:

$$\mathrm{Attn}^{(l)} = \mathrm{softmax}\left(\mathrm{mask}\left(\frac{\boldsymbol{Q}^{(l)}\boldsymbol{K}^{(l)T}}{\sqrt{d_k}}\right)\right), \quad \boldsymbol{X}^{\mathrm{qkv}(l)} = \mathrm{Attn}^{(l)}\boldsymbol{V}^{(l)}. \tag{17}$$

The output of the $l$-th attention layer is obtained as:

$$\boldsymbol{X}^{\mathrm{ao}(l)} = \mathrm{LN}\left(\boldsymbol{X}^{(l)} + \boldsymbol{X}^{\mathrm{qkv}(l)}\boldsymbol{W}^{O(l)}\right), \quad \boldsymbol{X}^{(l+1)} := \boldsymbol{X}^{\mathrm{do}(l)} = \mathrm{LN}\left(\mathrm{MLP}\left(\boldsymbol{X}^{\mathrm{ao}(l)}\right) + \boldsymbol{X}^{\mathrm{ao}(l)}\right), \tag{18}$$

where "LN" refers to Layer Normalization. The final output is obtained by projecting the output of the last layer $\boldsymbol{X}^{\mathrm{do}(L)}$ using a linear projection layer, followed by a softmax operation and argmax to obtain the predicted token.

## A.2. Experimental Setups

For those experiments about the Transformer structure, we train three Transformer models on a dataset of 200,000 samples, with each input sequence having a fixed length of 9 tokens. The vocabulary size $d_{\mathrm{vob}}$ is set to 200, and the model architecture includes an embedding dimension $d_m$ of 200, a feedforward dimension $d_f$ of 512, and query-key-value projection dimension $d_k$ of 64. The Transformer-based model uses 2 decoder layers with 1 attention head per layer. The training is conducted for 1000 epochs with a batch size of 100, and gradient clipping is applied with a maximum norm of 1. The AdamW optimizer is employed with an initial learning rate of $1 \times 10^{-5}$. The initialization rates of the three models are $\gamma = 0.3, 0.5, 0.8$.

For those experiments related to Emb-MLP, we train three Emb-MLP models with $d_{\mathrm{vob}} = 200, d_m = 200, d_f = 512$ and initialization scales $\gamma = 0.3, 0.5, 0.8$. A dataset comprising $1,000,000$ data pairs is employed. We set that $\mathcal{A}_{\mathrm{mem}} = \{1, 2, \cdots, 10\}, \mathcal{A}_{\mathrm{rsn}} = \{11, 12, \cdots, 20\}, \mathcal{Z} = \{21, 22, \cdots, 120\}, \mathcal{M} = \{(11, 13), (13, 11)\}$. The initial learning rate is $5 \times 10^{-6}$ and all other training setups remain consistent with those described in the first paragraph.

For Figure 1 and Figure 7, we use GPT-2 models with an initialization scale $\gamma = 0.3, 0.5, 0.8$. The dataset contains 10,000 data sequences, with half of them sourced from PrOntoQA and the other half from TinyStories. The AdamW optimizer is employed with an initial learning rate of $1 \times 10^{-5}$. The model is trained for 200 epochs, ensuring that the loss for both datasets decreases to a similar level.

# B. Theory Details

## B.1. Proof of Proposition 1

**Lemma 2.** *For any token $s$, let $\{(X^{s,i}, y^{s,i})\}_{i=1}^{n_s}$ denote all input sequences containing $s$ and corresponding labels. The gradient flow of $\boldsymbol{w}^{\mathrm{emb},s}$ can be expressed as:*

$$\frac{d\boldsymbol{w}^{\mathrm{emb},s}}{dt} = \frac{1}{n}\sum_{i=1}^{n_s}\left(\left(\boldsymbol{p}^{s,i} - \boldsymbol{y}^{s,i}\right)\boldsymbol{W}^{(2)T}\right) \odot \sigma'\left(\sum_{x \in X^{s,i}} \boldsymbol{w}^{\mathrm{emb},x}\boldsymbol{W}^{(1)}\right)\boldsymbol{W}^{(1)T}, \tag{19}$$

*where $\boldsymbol{p}^{s,i} = \mathrm{softmax}\left(\mathrm{G}_{\boldsymbol{\theta}}\left(\mathrm{X}^{s,i}\right)\right)$, $\sigma'$ denotes the derivative of $\sigma$ and $n$ means the count of all training data. $\odot$ represents the elements-wise production.*

*Proof.* For any data pair $\left(X^{s,i}, y^{s,i}\right)$, the cross entropy function $R$ could be expressed as:

$$R\left(X^{s,i}\right) = -\log \frac{e^{\boldsymbol{G_\theta}\left(X^{s,i}\right)_{y^{s,i}}}}{\sum_{j=1}^{d_{\text{vob}}} e^{\boldsymbol{G_\theta}(X^{s,i})_j}},$$

where the subscript $j$ represents the element index. Then the derivative of $R$ respect with $\boldsymbol{w}^{\text{emb},s}$ can be expressed as:

$$\frac{\partial R\left(X^{s,i}\right)}{\partial \boldsymbol{w}^{\text{emb},s}} = \sum_{j=1}^{d_{\text{vob}}} \frac{\partial R\left(X^{s,i}\right)}{\boldsymbol{G_\theta}\left(X^{s,i}\right)_j} \frac{\partial \boldsymbol{G_\theta}\left(X^{s,i}\right)_j}{\partial \boldsymbol{w}^{\text{emb},s}} = \sum_{j=1}^{d_{\text{vob}}} (\boldsymbol{p}_j^{s,i} - \boldsymbol{y}_j^{s,i}) \frac{\partial \boldsymbol{G_\theta}\left(X^{s,i}\right)_j}{\partial \boldsymbol{w}^{\text{emb},s}}.$$

Using the trace theorem, we obtain:

$$d\boldsymbol{G_\theta}\left(X^{s,i}\right)_j = \text{tr}\left(d\boldsymbol{G_\theta}\left(X^{s,i}\right)_j\right) = \text{tr}\left(d\sigma\left(\sum_{x\in X^{s,i}} \boldsymbol{w}^{\text{emb},x} \boldsymbol{W}^{(1)}\right) \boldsymbol{W}_{:,j}^{(2)}\right)$$

$$= \text{tr}\left(\sigma'\left(\sum_{x\in X^{s,i}} \boldsymbol{w}^{\text{emb},x} \boldsymbol{W}^{(1)}\right) \odot \left(d\boldsymbol{w}^{\text{emb},s} \boldsymbol{W}^{(1)}\right) \boldsymbol{W}_{:,j}^{(2)}\right)$$

$$= \text{tr}\left(\boldsymbol{W}^{(1)} \left(\boldsymbol{W}_{:,j}^{(2)} \odot \sigma'\left(\sum_{x\in X^{s,i}} \boldsymbol{w}^{\text{emb},x} \boldsymbol{W}^{(1)}\right)\right)^T d\boldsymbol{w}^{\text{emb},s}\right).$$

Then we have

$$\frac{\partial R\left(X^{s,i}\right)}{\partial \boldsymbol{w}^{\text{emb},s}} = \sum_{j=1}^{d_{\text{vob}}} \left(\boldsymbol{p}_j^{s,i} - \boldsymbol{y}_j^{s,i}\right) \left(\boldsymbol{W}_{:,j}^{(2)T} \odot \sigma'\left(\sum_{x\in X^{s,i}} \boldsymbol{w}^{\text{emb},x} \boldsymbol{W}^{(1)}\right)\right) \boldsymbol{W}^{(1)T}$$

$$= \left(\left(\boldsymbol{p}^{s,i} - \boldsymbol{y}^{s,i}\right) \boldsymbol{W}^{(2)T}\right) \odot \sigma'\left(\sum_{x\in X^{s,i}} \boldsymbol{w}^{\text{emb},x} \boldsymbol{W}^{(1)}\right) \boldsymbol{W}^{(1)T},$$

and furthermore

$$\frac{d\boldsymbol{w}^{\text{emb},s}}{dt} = -\frac{1}{n} \sum_{i=1}^{n_s} \frac{\partial R\left(X^{s,i}\right)}{\partial \boldsymbol{w}^{\text{emb},s}} = \frac{1}{n} \sum_{i=1}^{n_s} \left(\left(\boldsymbol{p}^{s,i} - \boldsymbol{y}^{s,i}\right) \boldsymbol{W}^{(2)T}\right) \odot \sigma'\left(\sum_{x\in X^{s,i}} \boldsymbol{w}^{\text{emb},x} \boldsymbol{W}^{(1)}\right) \boldsymbol{W}^{(1)T}.$$

$$\square$$

As the initialization scale decreases $\gamma \to 0$, with the Assumption 1, we have that $\sigma'\left(\sum_{x\in X^{s,i}} \boldsymbol{w}^{\text{emb},x} \boldsymbol{W}^{(1)}\right) = \boldsymbol{1}$, softmax $\left(\boldsymbol{G_\theta}\left(X^{s,i}\right)\right) = \frac{1}{d_{\text{vob}}} \boldsymbol{1}$, where $\boldsymbol{1} \in \mathbb{R}^{1 \times d_{\text{vob}}}$ means the vector with all elements equal to 1. Then the gradient flow of $\boldsymbol{w}^{\text{emb},s}$ could be approximated by the limit formulation, i.e.

$$\frac{d\boldsymbol{w}^{\text{emb},s}}{dt} = \frac{1}{n} \sum_{i=1}^{n_s} \left(\boldsymbol{y}^{s,i} - \frac{1}{d_{\text{vob}}} \boldsymbol{1}\right) \boldsymbol{W}^{(2)T} \boldsymbol{W}^{(1)T}. \tag{20}$$

Consider $n \to \infty$ and denote the random variable $Y^s$ which follows the distribution of $\{y^{s,i}\}_{i=1}^{n_s}$, then we obtain the asymptotic form by the law of large number

$$\frac{1}{n} \sum_{i=1}^{n_s} \boldsymbol{y}^{s,i} = r_s \mathbb{E}_{Y^s}\left[\boldsymbol{Y}^s\right] = r_s \boldsymbol{P}^s,$$

where $\boldsymbol{Y}^s$ is the one-hot representation of $Y^s$ and the $i$-th element of $\boldsymbol{P}^s$ is $\boldsymbol{P}_i^s = \mathbb{P}\left(Y^s = i\right)$. Then we obtain the Proposition 1.

## B.2. Distribution of $Y^s$

**Memory anchor.** Since we select a label for any key-anchor pair randomly from $\mathcal{U}(\mathcal{Z})$, the label $s$ meets would follow the same distribution for any $s \in \mathcal{A}_{\mathrm{mem}}$. specifically, we have

$$\mathbb{P}(Y^s = i) = \frac{1}{N_{\mathcal{Z}}} \delta_{i \in \mathcal{Z}}, \tag{21}$$

where $\delta_{i \in \mathcal{Z}} = 1$ if $i \in \mathcal{Z}$ otherwise 0.

**Reasoning anchor.** For any reasoning anchor $s \in \mathcal{A}_{\mathrm{rsn}}$, we assume that the other elements of a key-anchor pair containing $s$ is $z, a_1, a_2, \cdots, a_{q-1}$. Since the other elements are randomly chosen from the corresponding scope, the label could be represented as $y^s = s + z + \sum_{j=1}^{q-1} a_j$. Then $Y^s$ follows the distribution $s + Z + \sum_{j=1}^{q-1} A_j$, then we have

$$
\begin{aligned}
\mathbb{P}(Y^s = i) &= \mathbb{P}\left(Z + \sum_{j=1}^{q-1} A_j = i - s\right) \\
&= \sum_{\zeta=\zeta_{\min}}^{\zeta_{\max}} \mathbb{P}(Z = \zeta)\,\mathbb{P}\left(\sum_{j=1}^{q-1} A_j = i - s - \zeta\right) \\
&= \frac{1}{N_{\mathcal{Z}}} \frac{1}{N_{\mathcal{A}_{\mathrm{rsn}}}^{q-1}} \sum_{\zeta=\zeta_{\min}}^{\zeta_{\max}} \binom{q-1}{i-s-\zeta-(q-1)\alpha_{\min}^{\mathrm{rsn}}}_{N_{\mathcal{A}_{\mathrm{rsn}}}},
\end{aligned}
\tag{22}
$$

where the combination number $\binom{n}{j}_{k+1}$ can be defined by $\left(1 + x + \cdots + x^k\right)^n = \sum_{j=0}^{kn} \binom{n}{j}_{k+1} x^j$ (Caiado & Rathie, 2007). Specifically, when $q = 2$, we have the following result:

$$
\begin{aligned}
\mathbb{P}(Y^s = i) &= \sum_{\zeta=\zeta_{\min}}^{\zeta=\zeta_{\max}} \mathbb{P}(Z=\zeta)\,\mathbb{P}(A_1 = i - s - \zeta) \\
&= \begin{cases}
\sum_{\zeta=\zeta_{\min}}^{i-s-\alpha_{\min}^{\mathrm{rsn}}} \frac{1}{N_{\mathcal{Z}} N_{\mathcal{A}_{\mathrm{rsn}}}}, & i = \zeta_{\min} + \alpha_{\min}^{\mathrm{rsn}} + s, \cdots, \zeta_{\min} + \alpha_{\max}^{\mathrm{rsn}} + s. \\
\sum_{\zeta=i-s-\alpha_{\max}^{\mathrm{rsn}}}^{i-s-\alpha_{\min}^{\mathrm{rsn}}} \frac{1}{N_{\mathcal{Z}} N_{\mathcal{A}_{\mathrm{rsn}}}}, & i = \zeta_{\min} + \alpha_{\max}^{\mathrm{rsn}} + 1 + s, \cdots, \zeta_{\max} + \alpha_{\min}^{\mathrm{rsn}} + s. \\
\sum_{\zeta=i-s-\alpha_{\max}^{\mathrm{rsn}}}^{\zeta_{\max}} \frac{1}{N_{\mathcal{Z}} N_{\mathcal{A}_{\mathrm{rsn}}}}, & i = \zeta_{\max} + \alpha_{\min}^{\mathrm{rsn}} + 1 + s, \cdots, \zeta_{\max} + \alpha_{\max}^{\mathrm{rsn}} + s.
\end{cases} \\
&= \begin{cases}
\frac{i - s - \alpha_{\min}^{\mathrm{rsn}} - \zeta_{\min} + 1}{N_{\mathcal{Z}} N_{\mathcal{A}_{\mathrm{rsn}}}}, & i = \zeta_{\min} + \alpha_{\min}^{\mathrm{rsn}} + s, \cdots, \zeta_{\min} + \alpha_{\max}^{\mathrm{rsn}} + s. \\
\frac{1}{N_{\mathcal{Z}}}, & i = \zeta_{\min} + \alpha_{\max}^{\mathrm{rsn}} + 1 + s, \cdots, \zeta_{\max} + \alpha_{\min}^{\mathrm{rsn}} + s. \\
\frac{\zeta_{\max} + \alpha_{\max}^{\mathrm{rsn}} - i + s + 1}{N_{\mathcal{Z}} N_{\mathcal{A}_{\mathrm{rsn}}}}, & i = \zeta_{\max} + \alpha_{\min}^{\mathrm{rsn}} + 1 + s, \cdots, \zeta_{\max} + \alpha_{\max}^{\mathrm{rsn}} + s.
\end{cases}
\end{aligned}
$$

**Key.** For any $s \in \mathcal{Z}$, its labels come from two parts, memory mapping and reasoning mapping. In the memory mapping, $z$ meets each token in $\mathcal{Z}$ with the same probability $\frac{1}{N_{\mathcal{Z}}}$. In the reasoning mapping, the label $y^s = s + \sum_{i=1}^{q} a_i$. Assume that

the ratio of memory mapping is identified with reasoning mapping, then we have

$$\mathbb{P}(Y^s = i) = \frac{1}{2} \left( \mathbb{P}_{\text{mem}} \left( Y^s = i \right) + \mathbb{P}_{\text{rsn}} \left( Y^s = i \right) \right) \tag{23}$$

$$= \frac{1}{2} \left( \frac{1}{N_{\mathcal{Z}}} \delta_{i \in \mathcal{Z}} + \mathbb{P} \left( s + \sum_{j=1}^{q} A_j = i \right) \right) \tag{24}$$

$$= \frac{1}{2} \left( \frac{1}{N_{\mathcal{Z}}} \delta_{i \in \mathcal{Z}} + \frac{1}{N_{\mathcal{A}_{\text{rsn}}}^q} \left( i - s - q\alpha_{\min}^{\text{rsn}} \right)_{N_{\mathcal{A}_{\text{rsn}}}} \right). \tag{25}$$

Specifically, when $q = 2$

$$\mathbb{P}\left( Y^s = i \right) = \frac{1}{2} \left( \frac{1}{N_{\mathcal{Z}}} \delta_{i \in \mathcal{Z}} + \frac{N_{\mathcal{A}_{\text{rsn}}} - |\alpha_{\max}^{\text{rsn}} - \alpha_{\min}^{\text{rsn}} - i + s|}{N_{\mathcal{A}_{\text{rsn}}}^2} \right).$$

Generally, consider the usual sequence containing some noise. Then the label consists of a third part when $s$ appears as a noise. With a similar method, we have

$$\mathbb{P}_{noise} \left( Y^s = i \right) = \frac{1}{2} \left( \mathbb{P}_{noise,mem} \left( Y^s = i \right) + \mathbb{P}_{noise,rsn} \left( Y^s = i \right) \right)$$

$$= \frac{1}{2} \left( \frac{1}{N_{\mathcal{Z}}} \delta_{i \in \mathcal{Z}} + \mathbb{P} \left( Z + \sum_{j=1}^{q} A_j = i \right) \right)$$

$$= \frac{1}{2} \left( \frac{1}{N_{\mathcal{Z}}} \delta_{i \in \mathcal{Z}} + \frac{1}{N_{\mathcal{Z}}} \frac{1}{N_{\mathcal{A}_{\text{rsn}}}^q} \sum_{\zeta=\zeta_{\min}}^{\zeta_{\max}} \left( i - \zeta - q\alpha_{\min}^{\text{rsn}} \right)_{N_{\mathcal{A}_{\text{rsn}}}} \right).$$

Combine them together, we have in the general setting $\mathcal{X}^{(q,L)}$, we have that

$$\mathbb{P}(Y^s = i) = \frac{1}{2(L-q)} \left( \frac{1}{N_{\mathcal{Z}}} \delta_{i \in \mathcal{Z}} + \mathbb{P} \left( s + \sum_{j=1}^{q} A_j = i \right) \right) + \frac{L-q-1}{2(L-q)} \left( \frac{1}{N_{\mathcal{Z}}} \delta_{i \in \mathcal{Z}} + \mathbb{P} \left( Z + \sum_{j=1}^{q} A_j = i \right) \right)$$

$$= \frac{1}{2N_{\mathcal{Z}}} \delta_{i \in \mathcal{Z}} + \frac{1}{2(L-q)} \left( \mathbb{P} \left( s + \sum_{j=1}^{q} A_j = i \right) + (L-q-1) \mathbb{P} \left( Z + \sum_{j=1}^{q} A_j = i \right) \right)$$

$$= \frac{1}{2N_{\mathcal{Z}}} \delta_{i \in \mathcal{Z}} + \frac{1}{2(L-q) N_{\mathcal{A}_{\text{rsn}}}^q} \left( \left( i - s - q\alpha_{\min}^{\text{rsn}} \right)_{N_{\mathcal{A}_{\text{rsn}}}} + (L-q-1) \frac{1}{N_{\mathcal{Z}}} \sum_{\zeta=\zeta_{\min}}^{\zeta_{\max}} \left( i - \zeta - q\alpha_{\min}^{\text{rsn}} \right)_{N_{\mathcal{A}_{\text{rsn}}}} \right).$$

### B.3. Gradient Flow of Embedding Space in Emb-MLP

With the discussion in Section B.2, we obtain the detailed formulation of (5) for different anchors of different tasks. Specifically, we have the following result:

**Corollary 1.** *Given any $s \in \mathcal{A}_{\text{mem}}$, assume that $n \to \infty$ and assume the ratio of sequences containing $s$ in the training set $r_s$ keeps constant, then we have*

$$\frac{d\boldsymbol{w}^{\text{emb},s}}{dt} = r_s \left( \frac{\boldsymbol{\delta}^{\mathcal{Z}}}{N_{\mathcal{Z}}} - \frac{1}{d_{\text{vob}}} \boldsymbol{I} \right) \boldsymbol{W}^{(2)T} \boldsymbol{W}^{(1)T}, \tag{26}$$

*where $\boldsymbol{\delta}^{\mathcal{Z}} \in \mathbb{R}^d$ is a vector with elements equal to 1 for indices belonging to $\mathcal{Z}$, and 0 otherwise.*

**Corollary 2.** *Given any $s \in \mathcal{A}_{\text{rsn}}$, assume that $n \to \infty$ and the ratio of sequences containing $s$ in the training set $r_s$ remains constant. Then, the gradient flow of the embedding vector corresponding to $s$ is given by:*

$$\frac{d\boldsymbol{w}^{\text{emb},s}}{dt} = r_s \left( \boldsymbol{P}^s - \frac{1}{d_{\text{vob}}} \boldsymbol{I} \right) \boldsymbol{W}^{(2)T} \boldsymbol{W}^{(1)T}, \tag{27}$$

*where $\boldsymbol{P}^s \in \mathbb{R}^d$ is a probability vector whose $i$-th element is $\boldsymbol{P}_i^s = \frac{1}{N_{\mathcal{Z}}} \frac{1}{N_{\mathcal{A}_{\text{rsn}}}^{q-1}} \sum_{\zeta=\zeta_{\min}}^{\zeta_{\max}} \left( i - s - \zeta - (q-1)\alpha_{\min}^{\text{rsn}} \right)_{N_{\mathcal{A}_{\text{rsn}}}}$.*

## B.4. Proof of Lemma 1

*Proof.* We assume that $\boldsymbol{W}_{i,j}^{\text{emb},X} \sim \mathcal{N}\left(0, (d_m^{-\gamma})^2\right), \boldsymbol{W}_{i,j}^Q \sim \mathcal{N}\left(0, \left(d_k^{-\gamma}\right)^2\right), \boldsymbol{W}_{i,j}^K \sim \mathcal{N}\left(0, \left(d_k^{-\gamma}\right)^2\right)$. We have that

$$\left(\boldsymbol{W}^{\text{emb},X}\boldsymbol{W}^Q\boldsymbol{W}^{KT}\boldsymbol{W}^{\text{emb},X,T}\right)_{i,j} = \sum_{k=1}^{d_m}\sum_{l=1}^{d_m}\boldsymbol{W}_{i,k}^{\text{emb},X}\left(\sum_{p=1}^{d_k}\boldsymbol{W}_{k,p}^Q\boldsymbol{W}_{l,p}^K\right)\boldsymbol{W}_{j,l}^{\text{emb},X}$$

$$= \sum_{k=1}^{d_m}\sum_{l=1}^{d_m}\sum_{p=1}^{d_k}\boldsymbol{W}_{i,k}^{\text{emb},X}\boldsymbol{W}_{k,p}^Q\boldsymbol{W}_{l,p}^K\boldsymbol{W}_{j,l}^{\text{emb},X}$$

$$\sim \mathcal{N}\left(0, \left(\frac{d_m^2 d_k}{2\left(d_m^\gamma + d_k^\gamma\right)}\right)^2\right).$$

Then the attention operator

$$\frac{\left(\boldsymbol{W}^{\text{emb},X}\boldsymbol{W}^Q\boldsymbol{W}^{KT}\boldsymbol{W}^{\text{emb},X,T}\right)_{i,j}}{\sqrt{d_k}} \sim \mathcal{N}\left(0, \left(\frac{d_m^2\sqrt{d_k}}{2\left(d_m^\gamma + d_k^\gamma\right)}\right)^2\right).$$

Utilizing the Chebyshev's Inequality, then we have

$$\mathbb{P}\left(\frac{\left|\left(\boldsymbol{W}^{\text{emb},X}\boldsymbol{W}^Q\boldsymbol{W}^{KT}\boldsymbol{W}^{\text{emb},X,T}\right)_{i,j}\right|}{\sqrt{d_k}} > \delta\right) \le \frac{d_m^4 d_k}{4\delta^2\left(d_m^\gamma + d_k^\gamma\right)^2},$$

for any $\delta > 0$. Given any $\varepsilon \in (0,1]$, let $C = \frac{1}{2}\log_{d_m+d_k}\frac{d_m^4 d_k}{4\delta^2\varepsilon}$, then for any $\gamma > C$, we have

$$\mathbb{P}\left(\frac{\left|\left(\boldsymbol{W}^{\text{emb},X}\boldsymbol{W}_Q\boldsymbol{W}_K^T\boldsymbol{W}^{\text{emb},X,T}\right)_{i,j}\right|}{\sqrt{d_k}} > \delta\right) \le \frac{d_m^4 d_k}{4\delta^2\left(d_m^\gamma + d_k^\gamma\right)^2} \le \varepsilon,$$

which implies that $\boldsymbol{A}_{i,j} \xrightarrow{P} \frac{1}{i}$, for any $i \le j$ as $\gamma \to \infty$. $\qquad\square$

## B.5. Proof of Proposition 2, 3

For convenience in further analysis, we introduce the following notations $\boldsymbol{H}^{s,i} := (\overline{\boldsymbol{W}}^{\text{emb},X^{s,i}}\boldsymbol{W}^V\boldsymbol{W}^O + \boldsymbol{W}_L^{\text{emb},X^{s,i}})\boldsymbol{W}^{f1}, \boldsymbol{W}^{VO} = \boldsymbol{W}^V\boldsymbol{W}^O, \boldsymbol{W}^f = \boldsymbol{W}^{f1}\boldsymbol{W}^{f2}, \boldsymbol{p}^{s,i} = \text{softmax}(\boldsymbol{f}_{\boldsymbol{\theta}}(X^{s,i}))$. Firstly, we have the following result:

**Lemma 3.** *Given any token $s$, the gradient flow of $\boldsymbol{w}^{\text{emb},s}$ can be expressed as*

$$\frac{d\boldsymbol{w}^{\text{emb},s}}{dt} = -\frac{1}{n}\left(\sum_{i=1}^{n_s}\frac{1}{L}\left(\left(\boldsymbol{p}^{s,i} - \boldsymbol{y}^{s,i}\right)\boldsymbol{W}^{fT}\boldsymbol{W}^{VO,T} \odot \sigma'\left(\boldsymbol{H}^{s,i}\right)^T + \left(\boldsymbol{p}^{s,i} - \boldsymbol{y}^{s,i}\right)\boldsymbol{W}^{VO,T}\right)\right.$$

$$\left. + \sum_{i=1}^{\tilde{n}_s}\left(\boldsymbol{p}^{s,i} - \boldsymbol{y}^{s,i}\right)\boldsymbol{W}^{fT} \odot \sigma'\left(\boldsymbol{H}^{s,i}\right)^T + \left(\boldsymbol{p}^{s,i} - \boldsymbol{y}^{s,i}\right)\right),$$

*where $\tilde{n}_s$ denotes the time $s$ appears in the final position of a sequence.*

*Proof.* For any data pair $\left(X^{s,i}, y^{s,i}\right)$, we have

$$d\boldsymbol{f}_{\boldsymbol{\theta}}(X^{s,i})_j = d\left(\sigma\left(\boldsymbol{H}^{s,i}\right)\boldsymbol{W}_{:,j}^{f2} + \overline{\boldsymbol{W}}^{\text{emb},X^{s,i}}\boldsymbol{W}_{:,j}^{VO} + \boldsymbol{W}_{L,j}^{\text{emb},X^{s,i}}\right)$$

$$= \sigma'\left(\boldsymbol{H}^{s,i}\right) \odot \left(d\overline{\boldsymbol{W}}^{\text{emb},X^{s,i}}\boldsymbol{W}^{VO}\boldsymbol{W}^{f1} + d\boldsymbol{W}_L^{\text{emb},X^{s,i}}\boldsymbol{W}^{f1}\right)\boldsymbol{W}_{:,j}^{f2} + d\overline{\boldsymbol{W}}^{\text{emb},X^{s,i}}\boldsymbol{W}_{:,j}^{VO} + d\boldsymbol{W}_{L,j}^{\text{emb},X^{s,i}}.$$

By the trace theorem, we have

$$
\begin{aligned}
d\boldsymbol{f_\theta}\left(X^{s,i}\right)_j &= \mathrm{tr}\left(d\boldsymbol{f_\theta}\left(X^{s,i}\right)_j\right) \\
&= \mathrm{tr}\left(\boldsymbol{W}^{VO}\boldsymbol{W}^{f1}\left(\boldsymbol{W}^{f2}_{:,j}\odot\sigma'\left(\boldsymbol{H}^{s,i}\right)^T\right)d\overline{\boldsymbol{W}}^{\mathrm{emb},X^{s,i}}\right) + \mathrm{tr}\left(\boldsymbol{W}^{f1}\left(\boldsymbol{W}^{f2}_{:,j}\odot\sigma'\left(\boldsymbol{H}^{s,i}\right)^T\right)d\boldsymbol{W}^{\mathrm{emb},X^{s,i}}_{L,:}\right) \\
&\quad + \mathrm{tr}\left(\boldsymbol{W}^{VO}_{:,j}d\overline{\boldsymbol{W}}^{\mathrm{emb},X^{s,i}}\right) + \mathrm{tr}\left(d\boldsymbol{W}^{\mathrm{emb},X^{s,i}}_{L,j}\right) \\
&= \mathrm{tr}\left(\left(\boldsymbol{W}^{VO}\boldsymbol{W}^{f1}\left(\boldsymbol{W}^{f2}_{:,j}\odot\sigma'\left(\boldsymbol{H}^{s,i}\right)^T\right)+\boldsymbol{W}^{VO}_{:,j}\right)d\overline{\boldsymbol{W}}^{\mathrm{emb},X^{s,i}}\right) \\
&\quad + \mathrm{tr}\left(\left(\boldsymbol{W}^{f2}_{:,j}\odot\sigma'\left(\boldsymbol{H}^{s,i}\right)^T+\mathbf{1}\right)d\boldsymbol{W}^{\mathrm{emb},X^{s,i}}_{L,:}\right).
\end{aligned}
$$

Utilizing the chain rule, we have

$$
\begin{aligned}
\frac{\partial R\left(X^{s,i},y^{s,i}\right)}{\partial \boldsymbol{W}^{\mathrm{emb},s}} &= \sum_{j=1}^{d_m}\frac{\partial R\left(X^{s,i},y^{s,i}\right)}{\partial \boldsymbol{f_\theta}\left(X^{s,i}\right)_j}\frac{\partial \boldsymbol{f_\theta}\left(X^{s,i}\right)_j}{\partial \boldsymbol{W}^{\mathrm{emb},s}} = \sum_{j=1}^{d_m}\left(\boldsymbol{p}^{s,i}_j-\boldsymbol{y}^{s,i}_j\right)\frac{\partial \boldsymbol{f_\theta}\left(X^{s,i}\right)_j}{\partial \boldsymbol{W}^{\mathrm{emb},s}} \\
&= \begin{cases}
\frac{1}{L}\left(\left(\left(\boldsymbol{p}^{s,i}-\boldsymbol{y}^{s,i}\right)\boldsymbol{W}^{f2,T}\odot\sigma'\left(\boldsymbol{H}^{s,i}\right)\right)\boldsymbol{W}^{f1,T}\boldsymbol{W}^{VO,T}+\left(\boldsymbol{p}^{s,i}-\boldsymbol{y}^{s,i}\right)\boldsymbol{W}^{VO,T}\right) \\
\quad+\left(\left(\boldsymbol{p}^{s,i}-\boldsymbol{y}^{s,i}\right)\boldsymbol{W}^{f2,T}\odot\sigma'\left(\boldsymbol{H}^{s,i}\right)\right)\boldsymbol{W}^{f1,T}+\left(\boldsymbol{p}^{s,i}-\boldsymbol{y}^{s,i}\right), & s\text{ occurs on last position.} \\
\frac{1}{L}\left(\left(\left(\boldsymbol{p}^{s,i}-\boldsymbol{y}^{s,i}\right)\boldsymbol{W}^{f2,T}\odot\sigma'\left(\boldsymbol{H}^{s,i}\right)\right)\boldsymbol{W}^{f1,T}\boldsymbol{W}^{VO,T}+\left(\boldsymbol{p}^{s,i}-\boldsymbol{y}^{s,i}\right)\boldsymbol{W}^{VO,T}\right), & \text{otherwise.}
\end{cases}
\end{aligned}
$$

Then we obtain the gradient flow as

$$
\begin{aligned}
\frac{d\boldsymbol{w}^{\mathrm{emb},s}}{dt} &= -\frac{1}{n}\left(\sum_{i=1}^{n_s}\frac{1}{L}\left(\left(\boldsymbol{p}^{s,i}-\boldsymbol{y}^{s,i}\right)\boldsymbol{W}^{f2,T}\odot\sigma'\left(\boldsymbol{H}^{s,i}\right)\boldsymbol{W}^{f1,T}\boldsymbol{W}^{VO,T}+\left(\boldsymbol{p}^{s,i}-\boldsymbol{y}^{s,i}\right)\boldsymbol{W}^{VO,T}\right)\right. \\
&\quad\left.+\sum_{i=1}^{\tilde{n}_s}\left(\boldsymbol{p}^{s,i}-\boldsymbol{y}^{s,i}\right)\boldsymbol{W}^{f2,T}\odot\sigma'\left(\boldsymbol{H}^{s,i}\right)^T\boldsymbol{W}^{f1,T}+\left(\boldsymbol{p}^{s,i}-\boldsymbol{y}^{s,i}\right)\right).
\end{aligned}
$$

$\square$

As the initialization scales decrease to zero, we derive the gradient flow under a small initialization scale as follows via Assumption 1.

$$
\begin{aligned}
\frac{d\boldsymbol{w}^{\mathrm{emb},s}}{dt} &= \frac{1}{n}\left(\sum_{i=1}^{n_s}\frac{1}{L}\left(\boldsymbol{y}^{s,i}-\frac{1}{d_m}\mathbf{1}\right)\left(\boldsymbol{W}^{VO}\boldsymbol{W}^f+\boldsymbol{W}^{VO}\right)^T\right. \\
&\quad\left.+\sum_{i=1}^{\tilde{n}_s}\left(\boldsymbol{y}^{s,i}-\frac{1}{d_m}\mathbf{1}\right)\boldsymbol{W}^{f,T}+\left(\boldsymbol{y}^{s,i}-\frac{1}{d_m}\mathbf{1}\right)\right).
\end{aligned}
\tag{28}
$$

We consider the ideal condition $n\to\infty$, with the law of large number, (28) can be approximated as follow:

$$
\frac{d\boldsymbol{w}^{\mathrm{emb},s}}{dt} = \frac{r_s}{L}\mathbb{E}_{Y^s}\left[\boldsymbol{Y}^s-\frac{1}{d_m}\mathbf{1}\right]\tilde{\boldsymbol{W}}.
\tag{29}
$$

With the distribution we discussed in Section B.2, we complete the proof of Proposition 2, 3.

## B.6. Proof of Theorem 1

*Proof.* Consider the linear expansion of $\boldsymbol{w}^{\mathrm{emb},s} = \boldsymbol{w}_{t_0}^{\mathrm{emb},s} + \frac{d\boldsymbol{w}^{\mathrm{emb},s}}{dt}\eta$ where $\boldsymbol{w}_{t_0}^{\mathrm{emb},s}$ is the initialization of $\boldsymbol{w}^{\mathrm{emb},s}$, then we have

$$\boldsymbol{w}^{\mathrm{emb},s} = \boldsymbol{w}_{t_0}^{\mathrm{emb},s} + \frac{d\boldsymbol{w}^{\mathrm{emb},s}}{dt}\eta \tag{30}$$

$$= \boldsymbol{w}_{t_0}^{\mathrm{emb},s} + \frac{r_s\eta}{L}\mathbb{E}_{Y^s}\left[\boldsymbol{y}^s - \frac{1}{d_m}\mathbf{1}\right]\left(\boldsymbol{W}^{f,T} + \boldsymbol{I}\right)\left(\boldsymbol{W}^{VO,T} + \boldsymbol{I}\right) \tag{31}$$

$$= \frac{r_s\eta}{L}\mathbb{E}_{Y^s}\left[\boldsymbol{y}^s - \frac{1}{d_m}\mathbf{1}\right] + \boldsymbol{w}_{t_0}^{\mathrm{emb},s} + O\left(d_m^{-2\gamma}\mathbf{1}\right). \tag{32}$$

For any $s \in \mathcal{A}_{\mathrm{rsn}}$, the formulation can be rewritten as

$$\boldsymbol{w}_i^{\mathrm{emb},s} = \frac{r_s\eta}{L}\left(\mathbb{P}\left(s + Z + \sum_{j=1}^{q-1} A_j = i\right) - \frac{1}{d_m}\right) + \varepsilon, \tag{33}$$

where $\varepsilon \sim \mathcal{N}\left(0, (d_m^\gamma)^2\right)$. Let $q$ enlarge enough, then $\mathbb{P}\left(s + Z + \sum_{j=1}^{T-1} A_j = i\right)$ can be approximated by the following formulation using the Berry-Esseen central limit theorem

$$\sup_i \left|\mathbb{P}\left(s + Z + \sum_{j=1}^{q-1} = i\right) - \frac{1}{\sqrt{2\pi}\sigma_P}e^{-\frac{(i-s-\mu)^2}{2\sigma_P}}\right| \leq O\left(q^{-\frac{1}{2}}\right), \tag{34}$$

where $\mu$ and $\sigma_P$ is the expectation and standard deviation of $Z + \sum_{j=1}^{q-1} A_j$. Denote that $\tilde{\boldsymbol{w}}_i^{\mathrm{emb},s} = \frac{r_s\eta}{L}\left(\frac{1}{\sqrt{2\pi}\sigma_P}e^{-\frac{(i-s-\mu)^2}{2\sigma_P}} - \frac{1}{d_m}\right) + \varepsilon$, then we have:

$$\sup_i \left|\tilde{\boldsymbol{w}}_i^{\mathrm{emb},s} - \boldsymbol{w}_i^{\mathrm{emb},s}\right| \leq O\left(q^{-\frac{1}{2}} + d_m^{-\gamma}\right).$$

Then the difference in inner production can be derived as follows:

$$\sup_{i,j}\left|\left(\tilde{\boldsymbol{w}}^{\mathrm{emb},s_j}, \tilde{\boldsymbol{w}}^{\mathrm{emb},s_i}\right) - \left(\boldsymbol{w}^{\mathrm{emb},s_j}, \boldsymbol{w}^{\mathrm{emb},s_i}\right)\right| = \sup_{i,j}\left|\sum_k \tilde{\boldsymbol{w}}_k^{\mathrm{emb},s_j}\tilde{\boldsymbol{w}}_k^{\mathrm{emb},s_i} - \boldsymbol{w}_k^{\mathrm{emb},s_j}\boldsymbol{w}_k^{\mathrm{emb},s_i}\right|$$

$$\leq \sum_k \sup_{i,j}\left|\tilde{\boldsymbol{w}}_k^{\mathrm{emb},s_j}\tilde{\boldsymbol{w}}_k^{\mathrm{emb},s_i} - \boldsymbol{w}_k^{\mathrm{emb},s_j}\boldsymbol{w}_k^{\mathrm{emb},s_i}\right|$$

$$\leq O\left(d_m^{1-\gamma}\left(q^{-\frac{1}{2}} + d_m^{-\gamma}\right)\right).$$

Since an axis transformation does not affect the inner product, we set $\tilde{i} = i - \mu$, then we complete the proof of Theorem 1. □

## B.7. Validation of Theorem 1

To verify the validity and generality of our theoretical analysis, we compare the cosine similarity of the embedding vectors within the reasoning anchors between experimental results and theoretical approximations. The results in Figure 9 demonstrate that our theoretical estimates align well with the experimental results in most cases, with discrepancies observed only when $|s_i - s_j|$ becomes large, likely due to the omission of higher-order terms.

## B.8. Discussion about $W^V$

For the phenomenon where the $\boldsymbol{W}^V$ of the first attention module exhibits a preference for capturing the reasoning anchors, a general theoretical explanation would require a more comprehensive and sophisticated analysis. However, a similar result can be derived under a special and constrained condition.

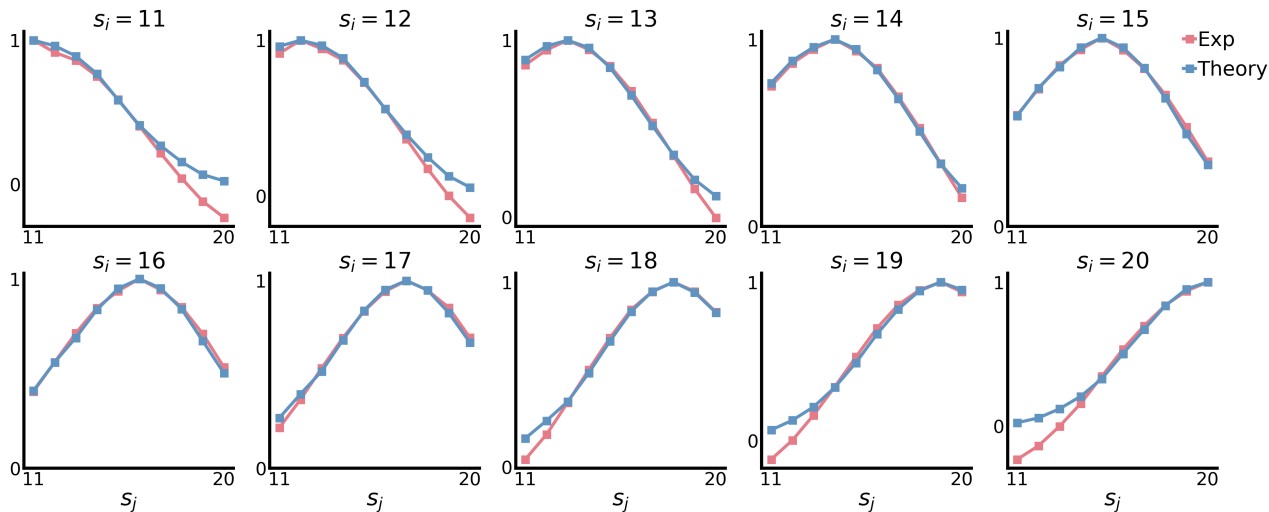

*Figure 9.* Cosine similarity comparison between experimental results $\cos\left(\boldsymbol{w}^{\mathrm{emb},s_i}, \boldsymbol{w}^{\mathrm{emb},s_j}\right)$ with theoretical approximations $\cos\left(\tilde{\boldsymbol{w}}^{\mathrm{emb},s_i}, \tilde{\boldsymbol{w}}^{\mathrm{emb},s_j}\right)$ (see (12)), for any $s_i, s_j \in \mathcal{A}_{\mathrm{rsn}}$.

**Theorem 2.** *Let $n, \gamma \to \infty$, $N_{\mathcal{Z}} = d_m$, Define that $A \sim \mathcal{U}\left(\mathcal{A}_{\mathrm{rsn}}\right)$ and $Y$ as a random variable which randomly takes value from the whole dataset's labels, then we have the following result:*

$$\frac{d\boldsymbol{W}^V}{dt} = \frac{1}{2}\mathbb{E}_A \boldsymbol{w}^{\mathrm{emb},A}\mathbb{E}_Y\left[\boldsymbol{Y} - \frac{1}{d_m}\mathbf{1}\right]^T \boldsymbol{W}^O\left(\boldsymbol{W}^f + I\right).$$

Theorem 2 highlights that $\boldsymbol{W}^V$ inherently demonstrates a preference for reasoning tasks, thereby enhancing its ability to capture information associated with reasoning anchors.

*Proof.* Firstly we have the following formulation:

**Lemma 4.** *Given the dataset $\{(X^i, y^i)\}_{i=1}^n$, the gradient flow of $\boldsymbol{W}^V$ can be expressed as follow:*

$$\frac{d\boldsymbol{W}^V}{dt} = -\frac{1}{n}\sum_{i=1}^n \left(\sigma'\left(\boldsymbol{H}^i\right) \odot \overline{\boldsymbol{W}}^{\mathrm{emb},X^i}\right)^T \left(\boldsymbol{p}^i - \boldsymbol{y}^i\right)\left(\boldsymbol{W}^O\boldsymbol{W}^f\right)^T + \overline{\boldsymbol{W}}^{\mathrm{emb},X^i,T}\left(\boldsymbol{p}^i - \boldsymbol{y}^i\right)\boldsymbol{W}^{O,T}. \quad (35)$$

*Proof.* For each data pair $\left(X^i, y^i\right)$, we have

$$\boldsymbol{f_\theta}\left(X^i\right)_j = \sigma\left(\left(\left(\boldsymbol{A}_{L,:}\boldsymbol{W}^{\mathrm{emb},X^i}\boldsymbol{W}^V\boldsymbol{W}^O + \boldsymbol{W}_{L,:}^{\mathrm{emb},X^i}\right)\boldsymbol{W}^{f1}\right)\boldsymbol{W}_{:,j}^{f2} + \boldsymbol{A}_{L,:}\boldsymbol{W}^{\mathrm{emb},X^i}\boldsymbol{W}^V\boldsymbol{W}_{:,j}^O + \boldsymbol{W}_{L,j}^{\mathrm{emb},X^i}.$$

Compute its differential, we have

$$d\boldsymbol{f_\theta}\left(\boldsymbol{W}^{\mathrm{emb},X^i}\right)_j = d\left(\sigma\left(\left(\left(\overline{\boldsymbol{W}}^{\mathrm{emb},X^i}\boldsymbol{W}^V\boldsymbol{W}^O + \boldsymbol{W}_{L,:}^{\mathrm{emb},X^i}\right)\boldsymbol{W}^{f1}\right)\boldsymbol{W}_{:,j}^{f2} + d\overline{\boldsymbol{W}}^{\mathrm{emb},X^i}\boldsymbol{W}^V\boldsymbol{W}_{:,j}^O\right)$$

$$= \sigma'\left(\boldsymbol{H}^i\right) \odot d\left(\left(\overline{\boldsymbol{W}}^{\mathrm{emb},X^i}\boldsymbol{W}^V\boldsymbol{W}^O + \boldsymbol{W}_{L,:}^{\mathrm{emb},X^i}\right)\boldsymbol{W}^{f1}\right)\boldsymbol{W}_{:,j}^{f2} + \overline{\boldsymbol{W}}^{\mathrm{emb},X^i}d\boldsymbol{W}^V\boldsymbol{W}_{:,j}^O$$

$$= \sigma'\left(\boldsymbol{H}^i\right) \odot \overline{\boldsymbol{W}}^{\mathrm{emb},X^i}d\boldsymbol{W}^V\boldsymbol{W}^O\boldsymbol{W}^{f1}\boldsymbol{W}_{:,j}^{f2} + \overline{\boldsymbol{W}}^{\mathrm{emb},X^i}d\boldsymbol{W}^V\boldsymbol{W}_{:,j}^O.$$

Using the trace theorem

$$d\boldsymbol{f_\theta}\left(X^i\right)_j = \operatorname{tr}\left(d\boldsymbol{f_\theta}\left(X^i\right)_j\right) = \operatorname{tr}\left(\sigma'\left(\boldsymbol{H}^i\right)\odot\overline{\boldsymbol{W}}^{\mathrm{emb},X^i}d\boldsymbol{W}^V\boldsymbol{W}^O\boldsymbol{W}^f_{:,j}\right) + \operatorname{tr}\left(\overline{\boldsymbol{W}}^{\mathrm{emb},X^i}d\boldsymbol{W}^V\boldsymbol{W}^O_{:,j}\right)$$

$$= \operatorname{tr}\left(\boldsymbol{W}^O\boldsymbol{W}^f_{:,j}\sigma'\left(\boldsymbol{H}^i\right)\odot\overline{\boldsymbol{W}}^{\mathrm{emb},X^i}d\boldsymbol{W}^V\right) + \operatorname{tr}\left(\boldsymbol{W}^O_{:,j}\overline{\boldsymbol{W}}^{\mathrm{emb},X^i}d\boldsymbol{W}^V\right)$$

$$= \operatorname{tr}\left(\left(\boldsymbol{W}^O\boldsymbol{W}^f_{:,j}\sigma'\left(\boldsymbol{H}^i\right)\odot\overline{\boldsymbol{W}}^{\mathrm{emb},X^i} + \boldsymbol{W}^O_{:,j}\overline{\boldsymbol{W}}^{\mathrm{emb},X^i}\right)d\boldsymbol{W}^V\right),$$

which suggests that

$$\frac{\partial\boldsymbol{f_\theta}\left(X^i\right)_j}{\partial\boldsymbol{W}^V} = \left(\boldsymbol{W}^O\boldsymbol{W}^f_{:,j}\sigma'\left(\boldsymbol{H}^i\right)\odot\overline{\boldsymbol{W}}^{\mathrm{emb},X^i} + \boldsymbol{W}^O_{:,j}\overline{\boldsymbol{W}}^{\mathrm{emb},X^i}\right)^T.$$

Utilizing the chain rule, we have

$$\frac{\partial R\left(X^i\right)}{\partial\boldsymbol{W}^V} = \sum_{j=1}^{d_m}\frac{\partial R\left(X^i\right)}{\partial\boldsymbol{f_\theta}\left(X^i\right)_j}\frac{\partial\boldsymbol{f_\theta}\left(X^i\right)_j}{\partial\boldsymbol{W}^V}$$

$$= \sum_{j=1}^{d_m}\left(\boldsymbol{p}^i_j - \boldsymbol{y}^i_j\right)\left(\boldsymbol{W}^O\boldsymbol{W}^f_{:,j}\sigma'\left(\boldsymbol{H}^i\right)\odot\overline{\boldsymbol{W}}^{\mathrm{emb},X^i} + \boldsymbol{W}^O_{:,j}\overline{\boldsymbol{W}}^{\mathrm{emb},X^i}\right)^T$$

$$= \left(\sigma'\left(\boldsymbol{H}^i\right)\odot\overline{\boldsymbol{W}}^{\mathrm{emb},X^i}\right)^T\left(\boldsymbol{p}^i - \boldsymbol{y}^i\right)\left(\boldsymbol{W}^O\boldsymbol{W}^f\right)^T + \overline{\boldsymbol{W}}^{\mathrm{emb},X^i,T}\left(\boldsymbol{p}^i - \boldsymbol{y}^i\right)\boldsymbol{W}^{O,T},$$

where $\boldsymbol{p}^i = \operatorname{softmax}(\boldsymbol{f_\theta}(X^i))$. Then gradient flow of $\boldsymbol{W}^V$ can be expressed as

$$\frac{d\boldsymbol{W}^V}{dt} = -\frac{1}{n}\sum_{i=1}^n\frac{\partial R\left(X^i\right)}{\partial\boldsymbol{W}^V}$$

$$= -\frac{1}{n}\sum_{i=1}^n\left(\sigma'\left(\boldsymbol{H}^i\right)\odot\overline{\boldsymbol{W}}^{\mathrm{emb},X^i}\right)^T\left(\boldsymbol{p}^i - \boldsymbol{y}^i\right)\left(\boldsymbol{W}^O\boldsymbol{W}^f\right)^T + \overline{\boldsymbol{W}}^{\mathrm{emb},X^i,T}\left(\boldsymbol{p}^i - \boldsymbol{y}^i\right)\boldsymbol{W}^{O,T}.$$

$\square$

When initialized with a small scale, the gradient flow for $\boldsymbol{W}^V$ could be interpreted by:

$$\frac{d\boldsymbol{W}^V}{dt} = \frac{1}{n}\sum_{i=1}^n\overline{\boldsymbol{W}}^{\mathrm{emb},X^i}\left(\boldsymbol{y}^i - \frac{1}{d_m}\mathbf{1}\right)^T\left(\boldsymbol{W}^O\boldsymbol{W}^f + \boldsymbol{W}^O\right)^T \tag{36}$$

$$= \frac{1}{nL}\sum_{i=1}^n\sum_{j=1}^L\boldsymbol{w}^{\mathrm{emb}}_{(i,j)}\left(\boldsymbol{y}^i - \frac{1}{d_m}\mathbf{1}\right)^T\left(\boldsymbol{W}^O\boldsymbol{W}^f + \boldsymbol{W}^O\right)^T, \tag{37}$$

where $\boldsymbol{w}^{\mathrm{emb}}_{(i,j)}$ denotes the $j$-th element of $\boldsymbol{W}^{\mathrm{emb},X^i}$. In this formulation, there are $nL$ tokens, and we reorder all tokens along with their corresponding labels. Let $\boldsymbol{w}^{\mathrm{emb},s_i}$ denote the embedding vector of the $i$-th token, and let $y^{s_i}$ be its corresponding label. Consequently, the gradient flow can be expressed as:

$$\frac{d\boldsymbol{W}^V}{dt} = \frac{1}{N}\sum_{i=1}^N\boldsymbol{w}^{\mathrm{emb},s_i}\left(\boldsymbol{y}^{s_i} - \frac{1}{d_m}\mathbf{1}\right)^T\left(\boldsymbol{W}^O\boldsymbol{W}^f + \boldsymbol{W}^O\right)^T. \tag{38}$$

If we interpret $\boldsymbol{w}^{\mathrm{emb},s_i}$ by its linear expansion $\boldsymbol{w}^{\mathrm{emb},s_i}_{t_0} + \eta\frac{d\boldsymbol{w}^{\mathrm{emb},s_i}}{dt}$, we obtain that

$$\frac{d\boldsymbol{W}^V}{dt} = \frac{1}{N}\sum_{i=1}^N\left(\boldsymbol{w}^{\mathrm{emb},s_i}_{t_0} + \eta\frac{d\boldsymbol{w}^{\mathrm{emb},s_i}}{dt}\right)\left(\boldsymbol{y}^{s_i} - \frac{1}{d_m}\mathbf{1}\right)^T\left(\boldsymbol{W}^O\boldsymbol{W}^f + \boldsymbol{W}^O\right)^T$$

$$= \frac{1}{N}\sum_{i=1}^N\left(\boldsymbol{w}^{\mathrm{emb},s_i}_{t_0} + \eta\frac{r_{s_i}}{L}\mathbb{E}_{Y^{s_i}}\left[\boldsymbol{Y}^{s_i} - \frac{1}{d_m}\mathbf{1}\right]\left(\boldsymbol{W}^{f,T} + \boldsymbol{I}\right)\left(\boldsymbol{W}^{VO,T} + \boldsymbol{I}\right)\right)\left(\boldsymbol{y}^{s_i} - \frac{1}{d_m}\mathbf{1}\right)^T\left(\boldsymbol{W}^O\boldsymbol{W}^f + \boldsymbol{W}^O\right)^T.$$

Let $\boldsymbol{W}^1 = \left(\left(\boldsymbol{W}^f\right)^T + \boldsymbol{I}\right)\left(\left(\boldsymbol{W}^{VO}\right)^T + \boldsymbol{I}\right), \boldsymbol{W}^2 = \left(\boldsymbol{W}^O\boldsymbol{W}^f + \boldsymbol{W}^O\right)^T$, then the formulation could be rewritten as

$$
\begin{aligned}
\frac{d\boldsymbol{W}^V}{dt} =& \frac{1}{N}\sum_{i=1}^{N}\left(\boldsymbol{w}_{t_0}^{\mathrm{emb},s_i} + \eta\frac{r_{s_i}}{L}\mathbb{E}_{Y^{s_i}}\left[\boldsymbol{Y}^{s_i} - \frac{1}{d_m}\mathbf{1}\right]\boldsymbol{W}^1\right)\left(\boldsymbol{y}^{s_i} - \frac{1}{d_m}\mathbf{1}\right)^T\boldsymbol{W}^2 \\
=& \mathbb{E}_{s_i,Y}\left[\left(\boldsymbol{w}_{t_0}^{\mathrm{emb},s_i} + \eta\frac{r_{s_i}}{L}\mathbb{E}_{Y^{s_i}}\left[\boldsymbol{Y}^{s_i} - \frac{1}{d_m}\mathbf{1}\right]\boldsymbol{W}^1\right)\left(\boldsymbol{y}^{s_i} - \frac{1}{d_m}\mathbf{1}\right)^T\boldsymbol{W}^2\right] \\
=& \mathbb{E}_{s_i,Y}\left[\eta\frac{r_{s_i}}{L}\mathbb{E}_{Y^{s_i}}\left[\boldsymbol{Y}^{s_i} - \frac{1}{d_m}\mathbf{1}\right]\boldsymbol{W}^1\left(\boldsymbol{Y}^{s_i} - \frac{1}{d_m}\mathbf{1}\right)^T\boldsymbol{W}^2\right] + \mathbb{E}_{s_i,Y^{s_i}}\left[\boldsymbol{w}_{t_0}^{\mathrm{emb},s_i}\left(\boldsymbol{Y}^{s_i} - \frac{1}{d_m}\mathbf{1}\right)^T\boldsymbol{W}^2\right] \\
=& \frac{r_s\eta}{L}\mathbb{E}_{s_i}\left[\mathbb{E}_{Y^{s_i}}\left[\boldsymbol{Y}^{s_i} - \frac{1}{d_m}\mathbf{1}\right]\boldsymbol{W}^1\mathbb{E}_{Y^{s_i}}\left[\boldsymbol{Y}^{s_i} - \frac{1}{d_m}\mathbf{1}\right]^T\boldsymbol{W}^2\right] + \mathbb{E}_{s_i,Y^{s_i}}\left[\boldsymbol{w}_{t_0}^{\mathrm{emb},s_i}\left(\boldsymbol{Y}^{s_i} - \frac{1}{d_m}\mathbf{1}\right)^T\boldsymbol{W}^2\right] \\
=& \frac{r_s\eta}{L}\mathbb{E}_Y\left[\boldsymbol{Y} - \frac{1}{d_m}\mathbf{1}\right]\boldsymbol{W}^1\mathbb{E}_Y\left[\boldsymbol{Y} - \frac{1}{d_m}\mathbf{1}\right]^T\boldsymbol{W}^2 + \mathbb{E}_{s_i,Y^{s_i}}\left[\boldsymbol{w}_{t_0}^{\mathrm{emb},s_i}\left(\boldsymbol{Y}^{s_i} - \frac{1}{d_m}\mathbf{1}\right)^T\boldsymbol{W}^2\right].
\end{aligned}
$$

While $\mathbb{E}_Y\left[\boldsymbol{Y} - \frac{1}{d_m}\mathbf{1}\right]_i = \mathbb{P}\left(Y = i\right) - \frac{1}{d_m}$, using the discussion in section B.2, Let $Z \sim \mathcal{U}\left(\mathcal{Z}\right), A_1, \cdots A_q \sim \mathcal{U}\left(\mathcal{A}_{\mathrm{rsn}}\right)$ we have

$$
\begin{aligned}
\mathbb{P}\left(Y = i\right) - \frac{1}{d_m} &= \frac{1}{2}\left(\mathbb{P}_{\mathrm{mem}}\left(Y = i\right) + \mathbb{P}_{\mathrm{rsn}}\left(Y = i\right)\right) - \frac{1}{d_m} \\
&= \frac{1}{2}\left(\frac{\delta_{i\in\mathcal{Z}}}{N_{\mathcal{Z}}} + \mathbb{P}\left(Z + \sum_{j=1}^{q}A_j = i\right)\right) - \frac{1}{d_m} \\
&= \frac{1}{2}\left(\mathbb{P}\left(Z + \sum_{j=1}^{q}A_j = i\right) - \frac{1}{d_m}\right) + \frac{1}{2}\left(\frac{\delta_{i\in\mathcal{Z}}}{N_{\mathcal{Z}}} - \frac{1}{d_m}\right) \\
&= \frac{1}{2}\left(\mathbb{E}_{A_1}\left[\mathbb{P}\left(Z + \sum_{j=1}^{q}A_j = i \mid A_1 = a\right)\right] - \frac{1}{d_m}\right) + \frac{1}{2}\left(\frac{\delta_{i\in\mathcal{Z}}}{N_{\mathcal{Z}}} - \frac{1}{d_m}\right) \\
&= \frac{1}{2}\mathbb{E}_{A_1}\left[\mathbb{P}\left(Z + \sum_{j=1}^{q}A_j = i | A_1 = a\right) - \frac{1}{d_m}\right] + \frac{1}{2}\left(\frac{\delta_{i\in\mathcal{Z}}}{N_{\mathcal{Z}}} - \frac{1}{d_m}\right) \\
&= \frac{1}{2}\mathbb{E}_{A_1}\left[\mathbb{E}_{Y^s}\left[\boldsymbol{y}^s - \frac{1}{d_m}\right]_i\right].
\end{aligned}
$$

Then we have that

$$
\begin{aligned}
\frac{d\boldsymbol{W}^V}{dt} &= \frac{r\eta}{2L}\mathbb{E}_{A_1}\left[\mathbb{E}_{Y^s}\left[\boldsymbol{Y}^s - \frac{1}{d_m}\mathbf{1}\right]\right]\boldsymbol{W}^1\mathbb{E}_Y\left[\boldsymbol{Y} - \frac{1}{d_m}\mathbf{1}\right]^T\boldsymbol{W}^2 + \mathbb{E}_{s_i,Y^{s_i}}\left[\boldsymbol{w}_{t_0}^{\mathrm{emb},s_i}\left(\boldsymbol{Y}^{s_i} - \frac{1}{d_m}\mathbf{1}\right)^T\boldsymbol{W}^2\right] \\
&= \frac{1}{2}\mathbb{E}_{A_1}\left[\boldsymbol{w}^{\mathrm{emb},A}\right]\mathbb{E}_Y\left[\boldsymbol{Y} - \frac{1}{d_m}\mathbf{1}\right]^T\boldsymbol{W}^2 + O\left(d_m^{-4\gamma}\mathbf{1}\right).
\end{aligned}
$$

$\square$

# C. Mechanisms under Varying Initialization Scales

## C.1. Embedding Space of Emb-MLP

Figure 10 exhibits the cosine similarity within the embedding space of Emb-MLP models with initialization rates $\gamma = 0.3$ and $\gamma = 0.5$. The results indicate that under a large initialization scale, the embedding space of the model becomes less influenced by the label distributions and instead relies predominantly on orthogonality to differentiate all tokens. This mechanism neglects the intrinsic relationships among tokens, leading to a loss of generalization.

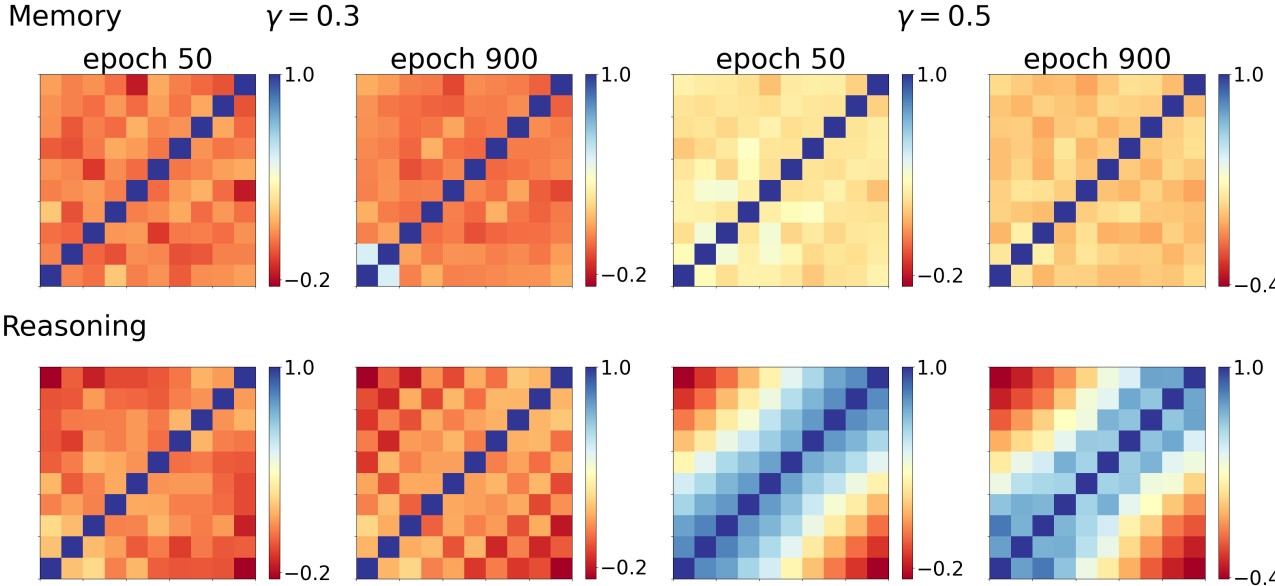

*Figure 10.* Cosine similarity among different anchors' embedding vectors of Emb-MLP under initialization rates $\gamma = 0.3, 0.5$ for memory anchors (top row) and reasoning anchors (bottom row).

## C.2. Embedding Space of Transformer

Figure 11 exhibits the structure of the Transformer's embedding space with $\gamma = 0.3$ and $\gamma = 0.5$. The left and middle panels exhibit the cosine similarity within the embedding space of memory anchors an reasoning anchors, demonstrating that a larger initialization scale promotes orthogonality among embedding vectors. The right panel presents the PCA projection of the embedding space, suggesting that under a large initialization scale, the embedding space lacks a meaningful structure conducive to learning the reasoning mapping. These findings suggest that a large initialization scale encourages differentiation of tokens primarily through orthogonality, taking tokens as independent from the others and neglecting intrinsic token relationships, and ultimately impairing the generalization capability.

## C.3. The First Attention Module of Transformer

Figure 12 exhibits the structure of the first attention module with $\gamma = 0.3$ and $\gamma = 0.5$ at epoch 200. The comparison reveals that a larger initialization scale results in a more complex attention mechanism, which exhibits no specific preference for any particular task.

## C.4. Low-rank Phenomena of Transformer

Figure 13 illustrates the distribution of singular value across different parameter matrices under varying initialization scales. The results reveal that as the initialization scale decreases, the parameter matrices exhibit a pronounced low-rank structure, which in turn facilitates a simpler learning mode.

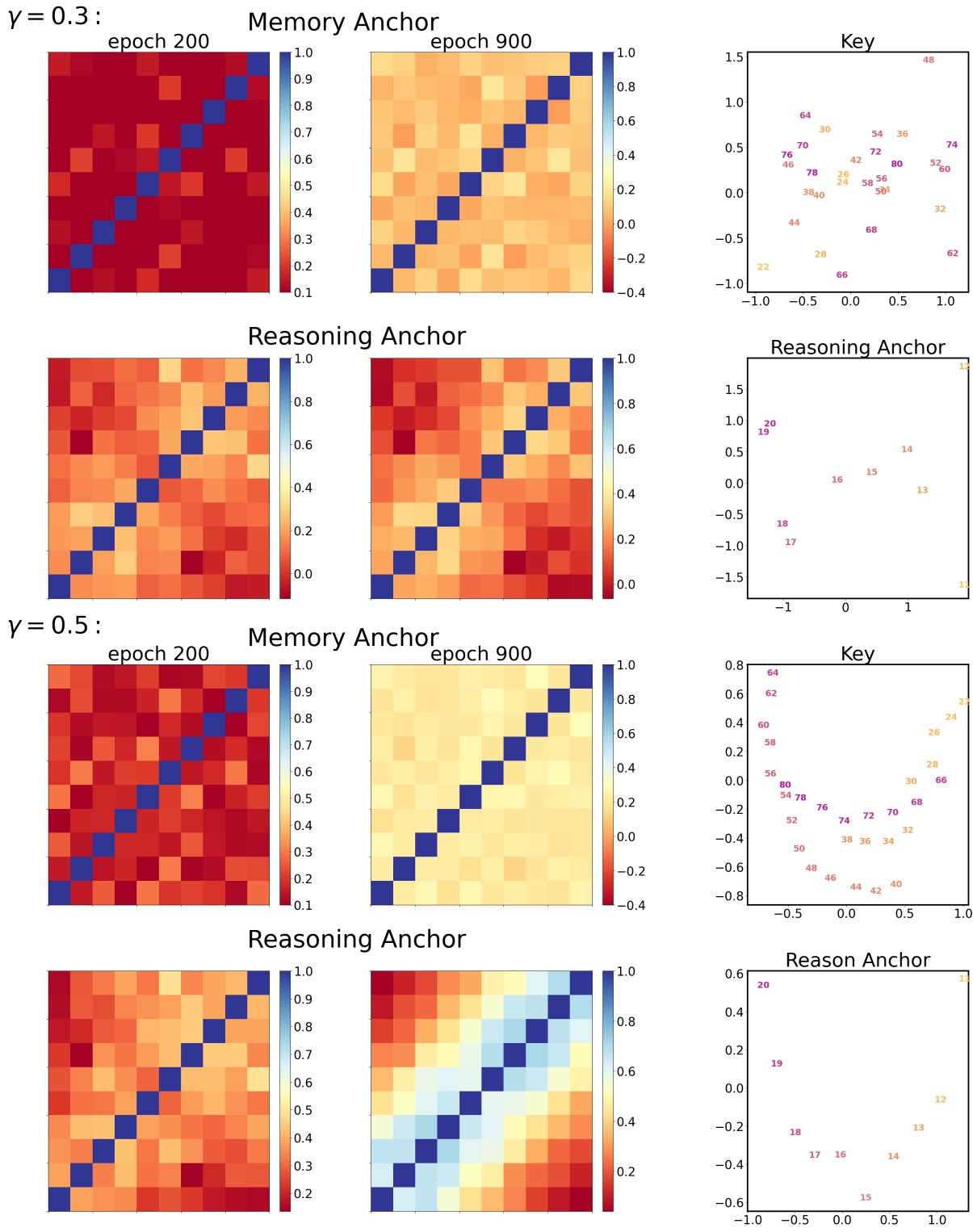

Figure 11. Characteristic of embedding space of Transformer with initialization rates $\gamma = 0.3, 0.5$. The left and middle panels depict the cosine similarity among embedding vectors of memory anchors and reasoning anchors at epochs 200 and 900. The right panel shows a PCA projection of the embedding space with the key and reasoning anchors.

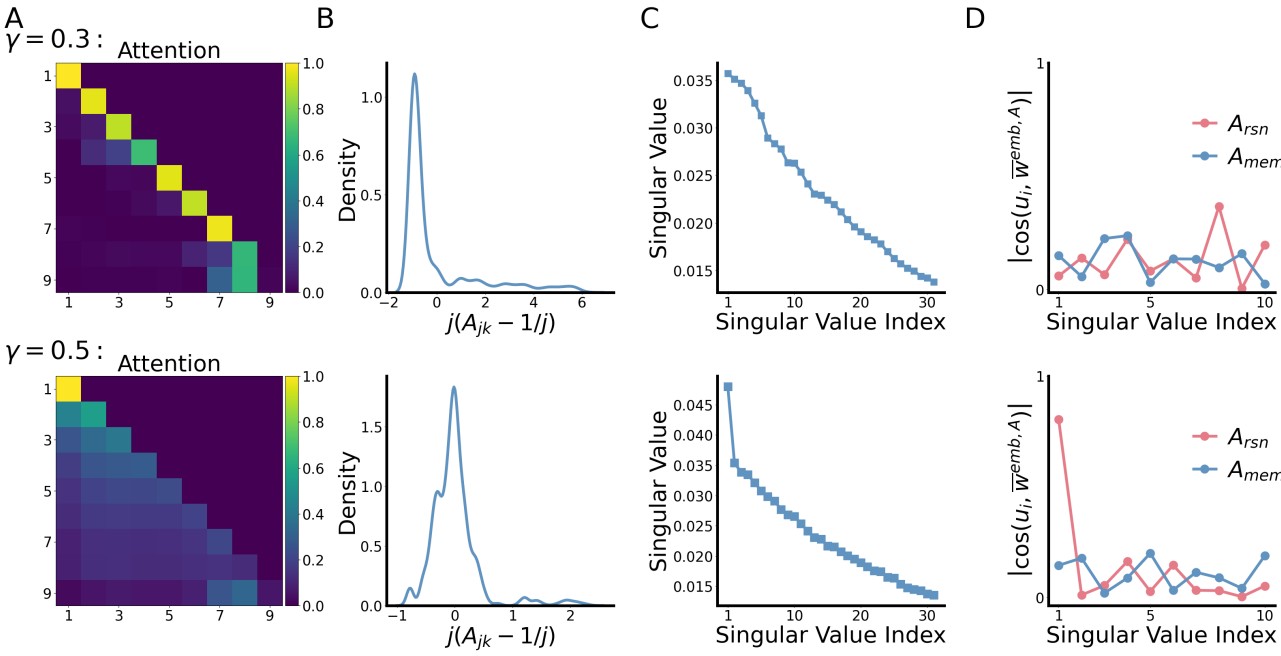

Figure 12. Characteristics of the first attention module of Transformers (step 200) with initialization rates $\gamma = 0.3$ (top row) and $\gamma = 0.5$ (bottom row). A: Heatmap of the attention matrix for a random sample. B: Distribution of the relative error between attention $A_{jk}$ and $\frac{1}{j}$ across all training sequences. C: Distribution of singular values of $\boldsymbol{W}^V$. D: Cosine similarity between the left singular vectors and average embedding vectors of the anchors.

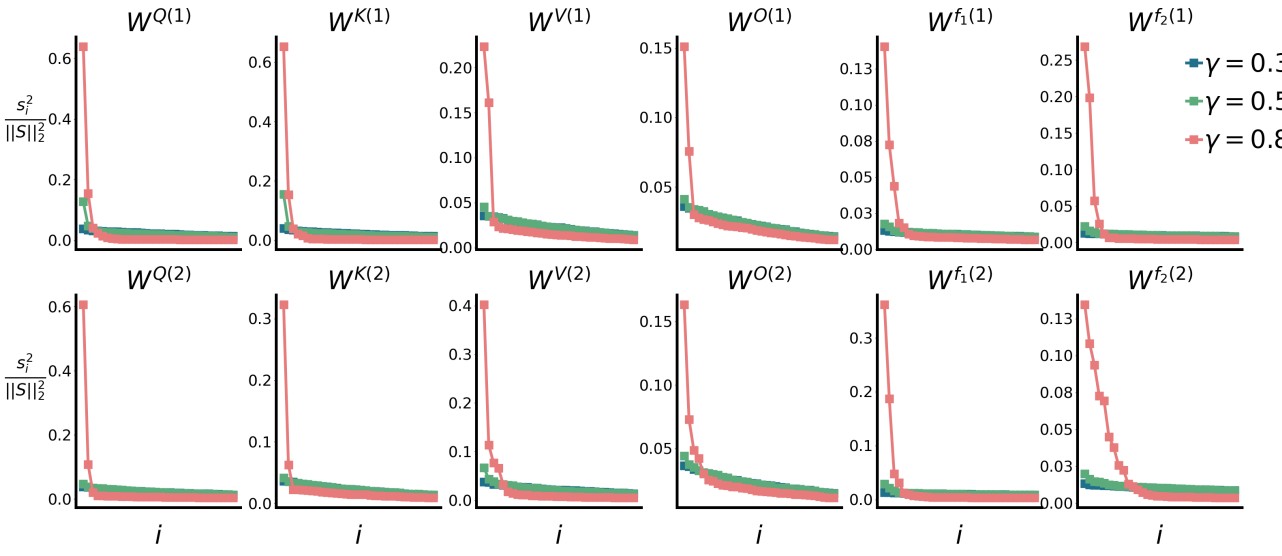

Figure 13. The distribution of singular value in different parameter matrices under different initialization scales (step 200). We denote the singular value vector by $S$ and the $i$-th largest singular value by $s_i$.

## C.5. Embedding Space in Real Language Tasks.

Figure 14 exhibits the cosine similarity within the embedding space of PrOntoQA and TinyStories tasks trained by GPT-2 models with initialization rates $\gamma = 0.3$ and $\gamma = 0.5$. It's noted that under a large initialization scale, the embedding vectors are mutually orthogonal, indicating that the model neglects the associations among different tokens.

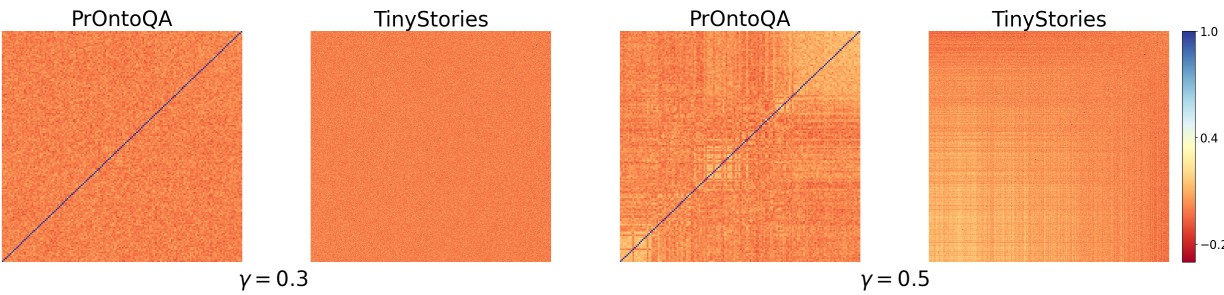

*Figure 14.* Characteristic of embedding space of PrOntoQA and TinyStories with initialization rates $\gamma = 0.3, 0.5$ (step 5000).

# D. The Second Attention Module

The function of the second attention module is to extract the key preceding the anchors $z_p$, and transfer its information to the last position. Figure 15A depicts the last row of the attention matrix before applying softmax, whose variation trend with respect to position index $i$ can be divided into three parts: (1) for $i \leq p$, the attention increases progressively as $i$ increases; (2) for $p + 1 \leq i \leq p + q$, the attention exhibits a slight decrease; and (3) for $i \geq p + q$, the attention drops sharply.

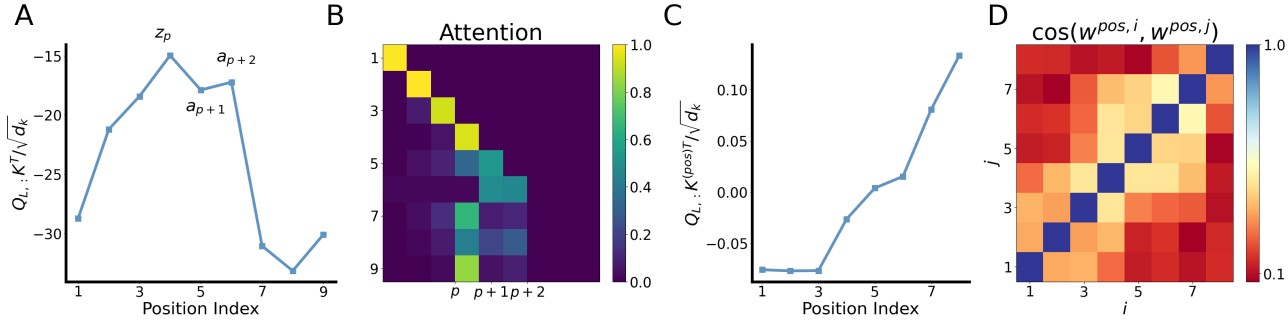

*Figure 15.* Characteristic of the second attention module. A: The last row of the second attention matrix (without applying softmax) for a randomly selected sequence. B: A heatmap of the second attention matrix for the same sequence. C: The last row of the matrix $\boldsymbol{Q}\boldsymbol{W}^{\text{pos}}\boldsymbol{W}^K/\sqrt{d_k}$ immediately before the final token. D: The cosine similarity of positional embeddings $\cos\left(\boldsymbol{w}^{\text{pos},i}, \boldsymbol{w}^{\text{pos},j}\right)$ for $i, j = 1, 2, \cdots, L - 1$.

Positional encoding plays a crucial role in this step. Figure 15C illustrates the last row of the attention matrix after substituting $\boldsymbol{K}$ with $\boldsymbol{W}^{\text{pos}}\boldsymbol{W}^K$, suggesting a increasing trend with the position index. Note that we only present the index immediately before the final token, as the key does not appear at the last position and thus is not required to adhere to the same pattern. Furthermore, since the reasoning anchor and the tokens following it are augmented with reasoning anchor's information in the first attention module, this information can be utilized to reduce the attention for tokens after the token. We construct a detailed mechanism about this in E.3.

# E. Reconstruction Mechanism for Information Capturing

To verify our observation is significant for information capturing for the Transformer model, we reconstruct the embedding space, the first attention module, and the second attention module and exhibit the process of extracting the key-anchor pair

from a reasoning sequence.

### E.1. Embedding Space

**Assumption 2** (Word Embedding). *We assume the embedding space has the following properties:*

1. $\cos\left(\boldsymbol{w}^{\mathrm{emb},s_{\mathrm{mem}}}, \boldsymbol{w}^{\mathrm{emb},s_{\mathrm{rsn}}}\right) = 0, \cos\left(\boldsymbol{w}^{\mathrm{emb},s_{\mathrm{rsn}}}, \boldsymbol{w}^{\mathrm{emb},s_{\mathrm{key}}}\right) = 0, \quad \forall s_{\mathrm{mem}} \in \mathcal{A}_{\mathrm{mem}}, s_{\mathrm{rsn}} \in \mathcal{A}_{\mathrm{rsn}}, s_{\mathrm{key}} \in \mathcal{Z}.$

2. *Fix any* $s_1 \in \mathcal{Z}$, $\cos\left(\boldsymbol{w}^{\mathrm{emb},s_1}, \boldsymbol{w}^{\mathrm{emb},s_2}\right) \geq \cos\left(\boldsymbol{w}^{\mathrm{emb},s_1}, \boldsymbol{w}^{\mathrm{emb},s_3}\right)$ *if* $|s_1 - s_2| \leq |s_1 - s_3|$, $\quad \forall s_2, s_3 \in \mathcal{Z}.$

3. *There exists a universal constant* $C_{\boldsymbol{w}} < \infty$ *such that* $||\boldsymbol{w}^{\mathrm{emb},s}||_\infty \leq C_{\boldsymbol{w}}$ *for any token s.*

In addition to word embeddings, position embeddings should be effectively utilized, as they play a critical role in the functionality of the second attention module. Here, we propose some assumptions about the relationship between word embeddings and position embeddings, with further characteristics to be elaborated upon later.

**Assumption 3** (Position Embedding 1). *Given any position embedding vector* $\boldsymbol{w}^{\mathrm{pos},i}$ *where i denotes the position index, we assume that* $\boldsymbol{w}^{\mathrm{pos},i} \perp \boldsymbol{w}^{\mathrm{emb},s}$ *for all* $i = 1, 2, \cdots L$ *and* $s \in \mathcal{Z} \cup \mathcal{A}_{\mathrm{rsn}} \cup \mathcal{A}_{\mathrm{mem}}.$

**Assumption 4** (Position Embedding 2). *We assume that* $\cos\left(\boldsymbol{w}^{\mathrm{pos},i}, \boldsymbol{w}^{\mathrm{pos},j}\right) = \cos\frac{|i-j|}{L}\pi$ *and* $||\boldsymbol{w}^{\mathrm{pos},i}|| = 1$ *for any* $i, j \in [1, L].$

Given any sequence $X$, the output of the embedding layer is

$$\boldsymbol{X}^{(1)} = e^X \boldsymbol{W}^{\mathrm{emb}} + \boldsymbol{W}^{\mathrm{pos}}.$$

### E.2. First Attention Module

In the first attention module, due to the impact of small initialization, the attention matrix $\boldsymbol{A}^{(1)}$ functions as an average operator. Specifically, the result of the first attention module can be interpreted as

$$\left(\mathrm{Attn}^{(1)}\left(\boldsymbol{X}^{(1)}\right)\boldsymbol{X}^{(1)}\boldsymbol{W}^{V(1)}\right)_j = \frac{1}{j}\left(\sum_{i \leq j} \boldsymbol{X}^{(1)}_{i,:}\right)\boldsymbol{W}^{V(1)}. \tag{39}$$

Furthermore, performing singular value decomposition (SVD) on the value projection matrix $\boldsymbol{W}^{V(1)}$ reveals that its largest singular value is significantly greater than the remaining singular values. The left singular vector corresponding to the largest singular value $\boldsymbol{W}^{V(1)}$ is highly similar to the embedding vectors of the reasoning anchors which indicates that $\boldsymbol{W}^{V(1)}$ can be approximated by

$$\boldsymbol{W}^{V(1)} = \lambda_V \left(\frac{1}{||\sum_{s \in \mathcal{A}_{\mathrm{rsn}}} \boldsymbol{W}^{\mathrm{emb},s}||_2} \sum_{s \in \mathcal{A}_{\mathrm{rsn}}} \boldsymbol{W}^{\mathrm{emb},s}\right)^T \boldsymbol{v}, \tag{40}$$

where $\lambda_V$ is the singular value and $\boldsymbol{v} \in \mathbb{R}^{1 \times d_k}$ denotes the right singular vector. Since $\boldsymbol{w}^{\mathrm{emb},s_{\mathrm{rsn}}} \perp \boldsymbol{w}^{\mathrm{emb},s_{\mathrm{mem}}}$ for any $s_{\mathrm{rsn}} \in \mathcal{A}_{\mathrm{rsn}}, s_{\mathrm{mem}} \in \mathcal{A}_{\mathrm{mem}}$ and $\boldsymbol{w}^{\mathrm{emb}} \perp \boldsymbol{w}^{\mathrm{pos}}$. Then the result of the attention operator can be interpreted as

$$\left(\mathrm{Attn}^{(1)}\left(\boldsymbol{X}^{(1)}\right)\boldsymbol{X}^{(1)}\boldsymbol{W}^{V(1)}\right)_{j,:} = \frac{\tilde{\lambda}_V}{j}\left(\sum_{i \leq j} \boldsymbol{W}^{\mathrm{emb},X}_{i,:}\right)\overline{\boldsymbol{w}}^{\mathrm{emb},T}_{\mathcal{A}_{\mathrm{rsn}}} \boldsymbol{v}, \tag{41}$$

where $\overline{\boldsymbol{w}}^{\mathrm{emb}}_{\mathcal{A}_{\mathrm{rsn}}} = \sum_{s \in \mathcal{A}_{\mathrm{rsn}}} \boldsymbol{w}^{\mathrm{emb},s}, \tilde{\lambda}_V = \frac{\lambda_V}{||\boldsymbol{w}^{\mathrm{emb}}_{\mathcal{A}_{\mathrm{rsn}}}||}$. Substituting the reasoning sequence $X^{\mathrm{rsn}}$ and memory sequence $X^{\mathrm{mem}}$, respectively, into this formulation, we derive the following results:

$$\left(\mathrm{Attn}^{(1)}\boldsymbol{X}^{\mathrm{mem},(1)}\boldsymbol{W}^{V(1)}\right)_{j,:} = 0, \tag{42}$$

$$\left(\mathrm{Attn}^{(1)}X^{\mathrm{rsn},(1)}\boldsymbol{W}^{V(1)}\right)_{j,:} = \begin{cases} \boldsymbol{0}, & j \leq p, \\ \frac{\tilde{\lambda}_V}{j}\left(\sum_{i=p+1}^{\min(j,p+q)} \boldsymbol{W}^{\mathrm{emb},X^{\mathrm{rsn}}}_i\right)\overline{\boldsymbol{w}}^{\mathrm{emb},T}_{\mathcal{A}_{\mathrm{rsn}}} \boldsymbol{v}, & p < j \leq L, \end{cases} \tag{43}$$

where $j$ means the row index. Thus, all tokens following the reasoning anchor are effectively "tagged," facilitating the identification of the anchor. Define the output of the first attention module is as follows:

$$\boldsymbol{X}^{(2)} = \boldsymbol{X}^{(1)} + \text{Attn}^{(1)}\left(\boldsymbol{X}^{(1)}\right)\boldsymbol{X}^{(1)}\boldsymbol{W}^{V(1)}.$$

Under the Assumption 2, we can formulate the output of the reasoning sequence further

$$\boldsymbol{X}^{\text{rsn},(2)}_{j,:} = \begin{cases} \boldsymbol{w}^{\text{emb},z_j} + \boldsymbol{w}^{\text{pos},j}, & j \leq p, \\[2mm] \boldsymbol{w}^{\text{emb},a_j} + \boldsymbol{w}^{\text{pos},j} + \dfrac{\tilde{\lambda}_V}{j}\left(\displaystyle\sum_{i=p+1}^{j} \boldsymbol{w}^{\text{emb},a_i}\right)\overline{\boldsymbol{w}}^{\text{emb},T}_{\mathcal{A}_{\text{rsn}}}\boldsymbol{v}, & p+1 \leq j \leq p+q, \\[4mm] \boldsymbol{w}^{\text{emb},z_j} + \boldsymbol{w}^{\text{pos},j} + \dfrac{\tilde{\lambda}_V}{j}\left(\displaystyle\sum_{i=p+1}^{p+q} \boldsymbol{w}^{\text{emb},a_i}\right)\overline{\boldsymbol{w}}^{\text{emb},T}_{\mathcal{A}_{\text{rsn}}}\boldsymbol{v}, & p+q+1 \leq j \leq L. \end{cases}$$

### E.3. Second Attention Module

We observe that the first attention module introduces additional information to all tokens with indices $j \geq p+1$. The subsequent challenge is to identify the reasoning tokens and the key token, and effectively propagate their information to the last position in the sequence. To achieve this, we construct the following attention distribution and demonstrate its properties:

**Definition 3** (Cliff Sequence). *Given a sequence $\boldsymbol{l} \in \mathbb{R}^L$, we define $\boldsymbol{l}$ as a $(p,q)$-cliff sequence if there exists $p, L \in \mathbb{N}^+$ such that $\boldsymbol{l}$ satisfies the following conditions:*

1. *(Increasing Segment) $\boldsymbol{l}_{i+1} > \boldsymbol{l}_i$ for all $i < p$.*

2. *(Plateau) $\frac{\boldsymbol{l}_{p-1}+\boldsymbol{l}_p}{2} \leq \boldsymbol{l}_{p+1}, \cdots, \boldsymbol{l}_{p+q} \leq \boldsymbol{l}_p$.*

3. *(Descending Segment) $\boldsymbol{l}_i < \boldsymbol{l}_1$ for all $p + q < i \leq L$.*

It is evident that if the attention of the last token forms a $(p,q)$-cliff sequence, it can effectively capture the information of the tokens and the key. Specifically, we have the following results to illustrate its feasibility.

**Lemma 5.** *For any $\varepsilon > 0$, there exists a $(p,q)$-cliff sequence $\boldsymbol{l}$ with norm $C$ such that $\text{softmax}(\boldsymbol{l})_i \leq \varepsilon$ for any $i \in [1, p-1] \cup [p+q+1, L]$.*

*Proof.* It's evident that we just need to illustrate $\text{softmax}(\boldsymbol{l})_{p-1} \to 0$ as $C \to \infty$. Denote that $\boldsymbol{l} = C\tilde{\boldsymbol{l}}$, then we have

$$\text{softmax}(\boldsymbol{l})_{p-1} = \frac{e^{C\tilde{l}_{p-1}}}{\sum_{j=1}^{L} e^{C\tilde{l}_j}}$$

$$= \frac{1}{\sum_{j\in[1,p-1]\cup[p+q+1,L]} e^{C(\tilde{l}_i - \tilde{l}_{p-1})} + \sum_{j\in[p,p+q]} e^{C(\tilde{l}_i - \tilde{l}_{p-1})}}.$$

Since that $\tilde{l}_j \leq \tilde{l}_{p-1}$ for any $j \in [1, p-1] \cup [p+q+1, L]$ and $\tilde{l}_j \geq \tilde{l}_{p-1}$ for any $j \in [p, p+q]$, so we have

$$\lim_{C\to\infty} \sum_{j\in[1,p-1]\cup[p+q+1,L]} e^{C(\tilde{l}_i - \tilde{l}_{p-1})} = 0 \quad \text{and} \quad \lim_{C\to\infty} \sum_{j\in[p,p+q]} e^{C(\tilde{l}_i - \tilde{l}_{p-1})} = \infty,$$

then $\text{softmax}(\text{l})_{\text{p}-1} \to 0$. $\qquad\square$

Here we provide a mechanism to construct a real matrix $\tilde{\boldsymbol{A}} \in \mathbb{R}^{d_m \times d_m}$ such that $\boldsymbol{X}^{\text{rsn},(2)}_{L,:}\tilde{\boldsymbol{A}}\boldsymbol{X}^{\text{rsn},(2),T}$ is a $(p,q)$-cliff sequence. Assume that $\tilde{\boldsymbol{A}} = \boldsymbol{\pi}\left(\text{span}\{\boldsymbol{w}^{pos}\}\right) - \mu\boldsymbol{v}^T\boldsymbol{v}, \mu > 0$, where $\boldsymbol{\pi}\left(\text{span}\{\boldsymbol{w}^{pos}\}\right)$ denotes the subspace spanned by

$\{\boldsymbol{w}^{pos}\}$. Then we have

$$
\boldsymbol{X}_{L,:}^{\mathrm{rsn},(2)}\tilde{\boldsymbol{A}}\boldsymbol{X}^{\mathrm{rsn},(2),T} = \begin{cases} \left(\boldsymbol{w}^{\mathrm{pos},L},\boldsymbol{w}^{\mathrm{pos},j}\right), & j \leq q, \\[2mm] \left(\boldsymbol{w}^{\mathrm{pos},L},\boldsymbol{w}^{\mathrm{pos},j}\right) - \dfrac{\tilde{\lambda}_V\mu}{L}\left(\displaystyle\sum_{i=p+1}^{p+q}\boldsymbol{w}^{\mathrm{emb},a_i},\overline{\boldsymbol{w}}_{\mathcal{A}_{\mathrm{rsn}}}^{\mathrm{emb}}\right)\left(\boldsymbol{v},\boldsymbol{w}^{\mathrm{emb},a_j}\right) \\[2mm] \quad -\dfrac{\tilde{\lambda}_V^2\mu}{jL}\left(\displaystyle\sum_{i=p+1}^{p+q}\boldsymbol{w}^{\mathrm{emb},a_i},\overline{\boldsymbol{w}}_{\mathcal{A}_{\mathrm{rsn}}}^{\mathrm{emb}}\right)\left(\displaystyle\sum_{i=p+1}^{j}\boldsymbol{w}^{\mathrm{emb},a_i},\overline{\boldsymbol{w}}_{\mathcal{A}_{\mathrm{rsn}}}^{\mathrm{emb}}\right), & p+1 \leq j \leq p+q, \\[2mm] \left(\boldsymbol{w}^{\mathrm{pos},L},\boldsymbol{w}^{\mathrm{pos},j}\right) - \dfrac{\tilde{\lambda}_V^2\mu}{jL}\left(\displaystyle\sum_{i=p+1}^{p+q}\boldsymbol{w}^{\mathrm{emb},a_i},\overline{\boldsymbol{w}}_{\mathcal{A}_{\mathrm{rsn}}}^{\mathrm{emb}}\right)^2, & p+q+1 \leq j \leq L. \end{cases}
$$

Define $\varphi_j = \left(\sum_{i=p+1}^{j}\boldsymbol{w}^{\mathrm{emb},a_i},\overline{\boldsymbol{w}}_{\mathcal{A}_{\mathrm{rsn}}}^{\mathrm{emb}}\right)$, applying the Assumption 4 on the position embedding, then we have the following result:

$$
\boldsymbol{X}_{L,:}^{\mathrm{rsn},(2)}\tilde{\boldsymbol{A}}\boldsymbol{X}^{\mathrm{rsn},(2),T} = \begin{cases} \cos\left(1-\dfrac{j}{L}\right)\pi, & j \leq p, \\[2mm] \cos\left(1-\dfrac{j}{L}\right)\pi - \dfrac{\tilde{\lambda}_V^2\mu}{jL}\varphi_{p+q}\left(\dfrac{j\|\boldsymbol{w}^{\mathrm{emb},a_j}\|}{\tilde{\lambda}_V}\cos\left(\boldsymbol{v},\boldsymbol{w}^{\mathrm{emb},a_j}\right)+\phi_j\right), & p+1 \leq j \leq p+q, \\[2mm] \cos\left(1-\dfrac{j}{L}\right)\pi - \dfrac{\tilde{\lambda}_V^2\mu}{jL}\varphi_{p+q}^2, & p+q+1 \leq j \leq L. \end{cases}
$$

To satisfy the Increasing Segment condition, we need that:

$$
\cos\left(1-\frac{j}{L}\right)\pi - \frac{\tilde{\lambda}_V^2\mu}{jL}\varphi_{p+q}\left(\frac{j\|\boldsymbol{w}^{\mathrm{emb},a_j}\|}{\tilde{\lambda}_V}\cos\left(\boldsymbol{v},\boldsymbol{w}^{\mathrm{emb},a_j}\right)+\varphi_j\right) \geq \frac{1}{2}\cos\left(1-\frac{p}{L}\right)\pi + \frac{1}{2}\cos\left(1-\frac{p-1}{L}\right)\pi,
$$

for any $p \in [1, L-q]$, $j \in [p+1, p+q]$. Denote that:

$$
\tilde{M} := \max_{p,q}\varphi_{p+q}, \qquad \tilde{m} := \min_{p,q}\varphi_{p+q}. \tag{44}
$$

Then we have

$$
\frac{\tilde{\lambda}_V^2\mu}{jL}\tilde{M}\left(\frac{j\|\boldsymbol{w}^{\mathrm{emb},a_j}\|}{\tilde{\lambda}_V\|\boldsymbol{v}\|}\cos\left(\boldsymbol{v},\boldsymbol{w}^{\mathrm{emb},a_j}\right)+\tilde{M}\right) \leq -\cos\left(\frac{p+1}{L}\right)\pi + \frac{1}{2}\cos\left(\frac{p}{L}\right)\pi + \frac{1}{2}\cos\left(\frac{p-1}{L}\right)\pi
$$

$$
\rightarrow \frac{\tilde{\lambda}_V^2\mu}{jL}\tilde{M}\left(\frac{j\|\boldsymbol{w}^{\mathrm{emb},a_j}\|}{\tilde{\lambda}_V\|\boldsymbol{v}\|}\cos\left(\boldsymbol{v},\boldsymbol{w}^{\mathrm{emb},a_j}\right)+\tilde{M}\right)
$$

$$
\leq \sqrt{\left(\frac{1}{2}\left(1-\cos\left(\frac{\pi}{L}\right)\right)\right)^2 + \left(\frac{3}{2}\sin\left(\frac{\pi}{L}\right)\right)^2}\cos\left(\frac{L-1}{L}\pi - \arctan\frac{3\sin\left(\frac{\pi}{L}\right)}{1-\cos\left(\frac{\pi}{L}\right)}\right).
$$

Denote the right side by $C_M$, and simplify it with

$$
\frac{\|\boldsymbol{w}^{\mathrm{emb},a_j}\|}{\tilde{\lambda}_V}\cos\left(\boldsymbol{v},\boldsymbol{w}^{\mathrm{emb},a_j}\right) \leq \frac{LC_M}{\tilde{\lambda}_V^2\mu\tilde{M}} - \frac{\tilde{M}}{L}. \tag{45}
$$

For another side, we assume that:

$$
\cos\left(1-\frac{j}{L}\right)\pi - \frac{\tilde{\lambda}_V^2\mu}{jL}\varphi_{p+q}\left(\frac{j\|\boldsymbol{w}^{\mathrm{emb},a_j}\|}{\tilde{\lambda}_V}\cos\left(\boldsymbol{v},\boldsymbol{w}^{\mathrm{emb},a_j}\right)+\varphi_j\right) \leq \cos\left(1-\frac{p}{L}\right)\pi,
$$

which implies that:

$$-\frac{\tilde{\lambda}_V^2\mu}{jL}\varphi_{p+q}\left(\frac{j||\boldsymbol{w}^{\mathrm{emb},a_j}||}{\tilde{\lambda}_V}\cos\left(\boldsymbol{v},\boldsymbol{w}^{\mathrm{emb},a_j}\right)+\varphi_j\right) \leq -2\sin\left(\frac{2p+q}{2L}\pi\right)\sin\left(\frac{q}{2L}\pi\right).$$

We have that:

$$\frac{||\boldsymbol{w}^{\mathrm{emb},a_j}||}{\tilde{\lambda}_V}\cos\left(\boldsymbol{v},\boldsymbol{w}^{\mathrm{emb},a_j}\right) \geq \frac{LC_m}{\tilde{\lambda}_V^2\mu\tilde{m}}-\frac{\tilde{m}}{L}. \tag{46}$$

These two conditions give the direction scope of $\boldsymbol{v}$. For the Descending Segment condition, we have that

$$\cos\left(1-\frac{1}{L}\right)\pi > \cos\left(1-\frac{j}{L}\right)\pi-\frac{\tilde{\lambda}_V^2\mu}{jL}\varphi_{p+q}^2$$
$$\rightarrow\tilde{\lambda}_V^2\mu > jL\left(\cos\left(\frac{\pi}{L}\right)-\cos\left(\frac{j\pi}{L}\right)\right)\varphi_{p+q}^{-2} \tag{47}$$
$$\rightarrow\tilde{\lambda}_V^2\mu > L^2\left(1+\cos\left(\frac{\pi}{L}\right)\right)\tilde{m}^{-2}.$$

With (45),(46), and (47), we could give a range of $\tilde{\lambda}_V,\mu$ and the direction of $\boldsymbol{v}$ which makes $\boldsymbol{X}_{L,:}^{\mathrm{rsn},(2)}\tilde{\boldsymbol{A}}\boldsymbol{X}^{\mathrm{rsn},(2),T}$ is a $(p,q)$-cliff sequence.

## F. Layer Normalization

We conduct an experiment with removing the Layer Normalization module, exhibiting the same phenomena, i.e., smaller initialization scales bias reasoning task, and results are depicted in Figure 16.

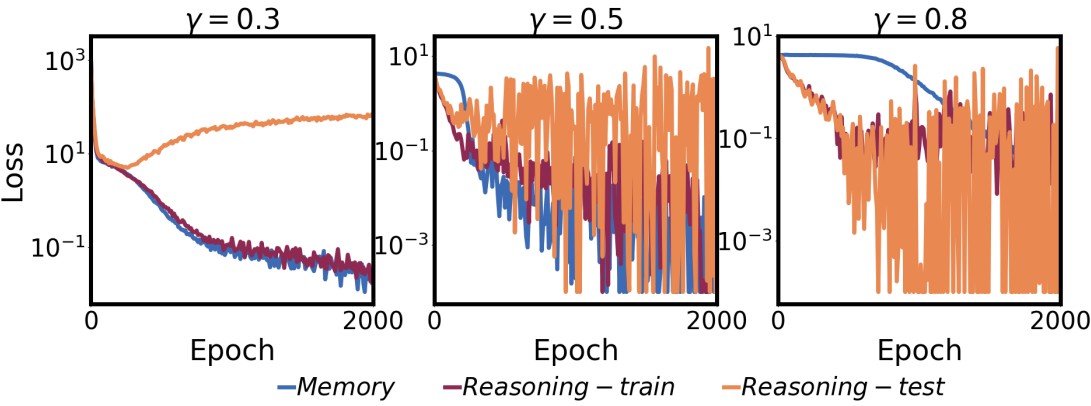

*Figure 16.* Training dynamics of Transformers under $\gamma = 0.3, 0.5, 0.8$ without Layer Normalization.

## G. Learning Rate

We conduct experiments with the learning rate belonging to $\left[10^{-5}, 5\times 10^{-4}\right]$. Figure 17 exhibits the loss dynamics under different $\gamma$, remaining consistent learning bias across these learning rate configurations. However, when the learning rate increases to 0.001, the training becomes highly unstable, manifesting a severe loss spike.

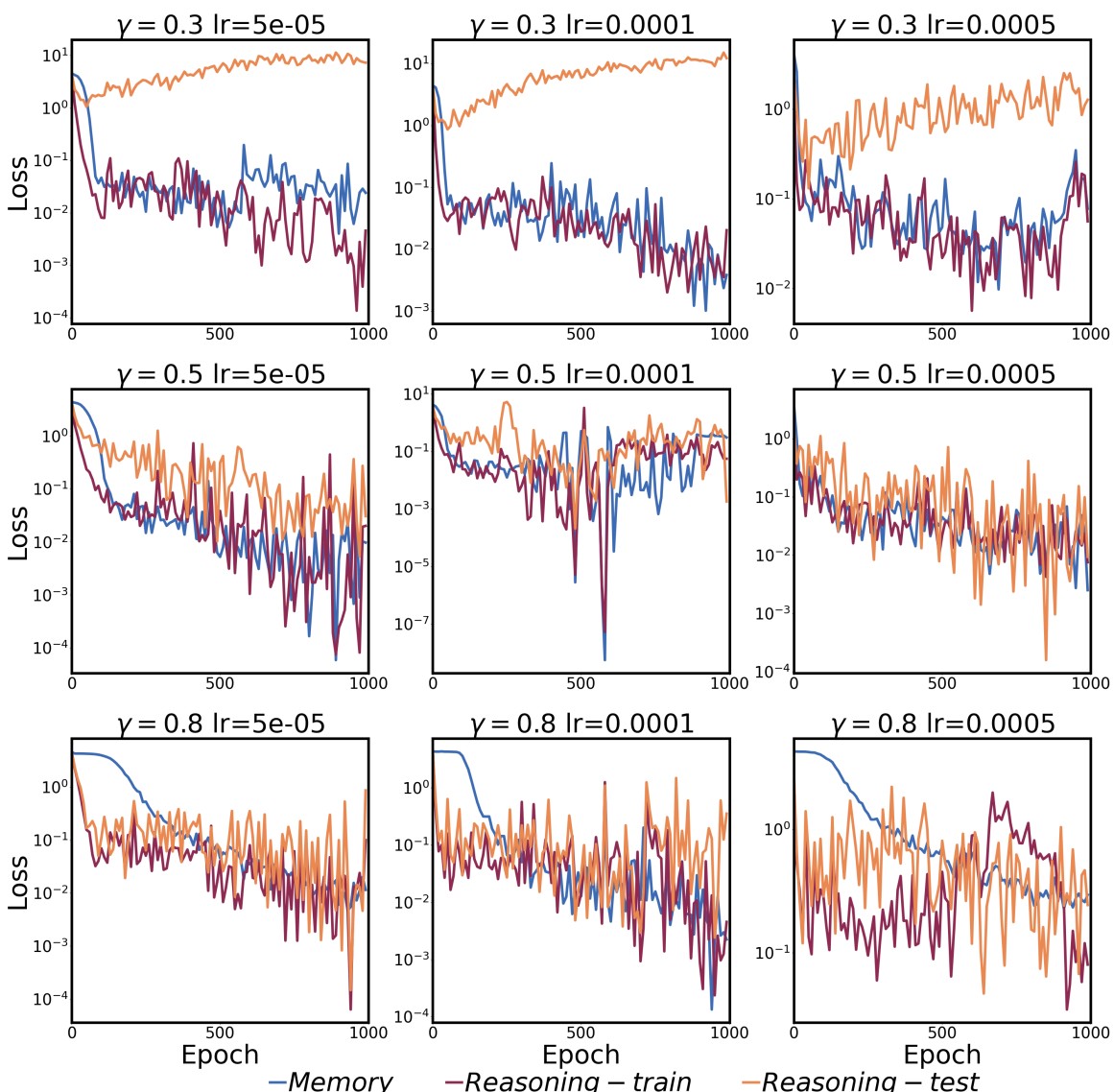

*Figure 17.* Training dynamics of Transformers under $\gamma = 0.3, 0.5, 0.8$ and varying learning rates.

