# OpenReview forum: "An Analysis for Reasoning Bias of Language Models with Small Initialization"
_ICML.cc/2025/Conference — ICML 2025 spotlightposter_

### Official Review · Reviewer_e1DG · 2025-03-08

**Overall Recommendation:** 4

**Summary:**

This paper aims to investigate how different initialization scales (small vs. large) may affect transformer models' ability to learn in different tasks (specifically in reasoning and memorization). The study pretrains a GPT-2 model with separately from smaller to larger initialization scales on reasoning and memorization tasks, where the datasets are both synthetic and real. The study finds an explicit bias that a smaller initialization scale of the transformer model can learn reasoning tasks much faster and generalizes better, while a larger initialization scale would make learning to memorize faster, but also reasoning worse. The authors used a variety of approaches to investigate the mechanisms of such bias. Specifically:

1. Theoretical analysis. The authors apply an Emb-MLP Model to analyze gradient dynamics as well as empirical representation evolutions. Results show that smaller initialization can amplify the gradient differences based on structured label distributions of reasoning tasks. Moreover, reasoning anchors quickly differentiate themselves while memory mappings do not. They also confirm this theoretical proposition by finding similar results from empirical results.

2. Transformer model analysis. The authors analyzed transformer embeddings and found that the structured clusters quickly form in the training of reasoning tasks. The authors also analyzed the attention matrices and proposed that the smaller initialization makes attention approximate the averaging operations.

Overall, this paper discovered that smaller initialization scales of transformer models can be good for reasoning and investigated the mechanisms underlying this discovery.

**Claims And Evidence:**

The authors claim that smaller initialization scale of transformer models can be good for learning reasoning tasks, while larger initialization scale can be better for memorization tasks. In the experiments, results did show that with smaller initialization, the transformer models learn faster and generalize better in the training of reasoning tasks, while larger scale initilization makes reasoning generalize worse but memorization faster.

**Essential References Not Discussed:**

I am not aware of any essential reference omitted by the authors.

**Experimental Designs Or Analyses:**

The analysis goes thoroughly from synthetic dataset to empirical dataset, from representation findings to theoretical analysis, which are robust and comprehensive.

**Methods And Evaluation Criteria:**

The study uses two styles of task and datasets to evaluate the training of reasoning and memorization. One is a synthetic compositional task and random numbered tasks. They are all sequences of numerical tokens and designed in the same order. However, the compositional task's output is an arithmetic mapping (e.g., addition) of keys while the memorization task's output only requires reproduction of the keys. This design has minimal differences of token distribution while requiring differences in the computation. Another type of datasets are benchmark datasets used for reasoning and mapping tasks. The authors analyze embeddings of models trained on these two datasets and found the former one exhibits significant hierarchical structure of logic representation in the reasoning dataset, while the latter one does not.

Overall, the authors used both empirical and synthetic datasets to evaluate the model's learning in reasoning and memorization, which are robust and thorough.

**Other Comments Or Suggestions:**

No other comments.

**Other Strengths And Weaknesses:**

The paper's visualization as well as structure is neat and easy to understand.

**Questions For Authors:**

Though may be out of the paper's scope, I still wonder in a typical pre-training setting, we used simply next token prediction to calculate the loss and optimize. I was wondering if this small initialization effect can generalize in reinforcement learning algorithms such as GRPO or PPO. The authors are not required to supplement any experiments. Sharing intuitions and insights into this topic would be interesting.

**Relation To Broader Scientific Literature:**

The paper proposes an important factor - the size of initialization of pre-training of a transformer-based language model in the reasoning and memorization task. This paper provides critical empirical and theoretical evidence that smaller initialization can work faster and better for the learning of reasoning models. This can be a both practical and fundamental concern of building better reasoning models, as well as understanding how reasoning models can work/fail to work.

**Theoretical Claims:**

Based on my limited expertise and understanding of transformer computations, I did not find any explicit incorrect proof.

---

> ### Author Rebuttal · Authors · 2025-03-31
>
> We sincerely appreciate your thorough review of our work and your valuable feedback and questions. Your recognition and support for our research have been immensely encouraging. We address your concerns as follows.
>
> **Question**: I was wondering if this small initialization effect can generalize in reinforcement learning algorithms such as GRPO or PPO.
>
> **Response**: It is important to study how does initialization scale impact the training behavior in reinforcement learning (RL). We provide some preliminary intuitions:
>
> 1.	The core of small initialization lies in reducing model complexity, compelling the pre-trained model to fit datasets through simpler, more generalizable patterns. With small initialization, a model can learn more reasoning patterns instead of memory patterns. RL can play a critical role in enhancing the reasoning patterns which are learned in the pre-training stage. On the other hand, employing small initialization for training reward model (RM) might improve the generalizability of RM scoring since small initialization could reduce overfitting to training data noise.
>
> 2.	Building on the complexity reduction principle, other complexity-reduction techniques such as larger weight decay could be integrated into the RL stage to further enhance generalization.
>
>
> 3.	In some post-training workflows, parameters of the pre-trained model are directly updated, rendering the concept of initialization irrelevant. However, in specialized approaches like LoRA (Low-Rank Adaptation), new update parameters are introduced and trained independently. As discussed in our response to Reviewer a4C, the initialization of these update parameters significantly influences fine-tuning results. Analogously, in RL settings, an update policy could be designed. Let $\pi=\pi_{pre-train}+\Delta \pi$ where $\Delta \pi $ is the trainable update policy.  Then the varying initialization schemes for $\Delta \pi$ might lead to distinct training behaviors.
>
> We hope our response could provide clarity and value to your question. Once again, we deeply appreciate your support and constructive feedback for our work.

---

> > ### Comment · Reviewer_e1DG · 2025-04-03
> >
> > Thanks for the rebuttal from the authors. The authors have well addressed my concerns and I would maintain my current score as acceptance.

---

> > > ### Author Response · Authors · 2025-04-03
> > >
> > > Dear Reviewer e1DG,
> > >
> > > Thank you for your detailed feedback and for taking the time to review our work. We greatly appreciate your recognition of our efforts, and we are pleased that our responses have addressed your concerns.
> > >
> > > The Authors.

---

### Official Review · Reviewer_FCaa · 2025-03-09

**Overall Recommendation:** 4

**Summary:**

The paper investigates initialization scale as one of the driving factors of bias towards different types of tasks. The paper considers two types of tasks:
1. Reasoning - represented by a sum over a key and its anchors given a key. This task is constructed to capture logical/arithmetic relationships, which requires generalization (And can be measured with the test set the authors provide)
2. Memory - represented by retrieving a predetermined random sample corresponding to a key-anchor pair. The memory mapping is entirely random, which means. ithcannot be inferred, only memorized.
There are also noise tokens provided to increase the difficulty of the learning problem/control the signal to noise.

The impact of initialization scale $\gamma$  (weights are initialized $W\in\mathbb R^{d_1\times d_2}, W_{i,j}\sim\mathcal N(\mu=0,\sigma=d_2^{-\gamma})$ )  upon the relative performance of 1. and 2. is investigated.

It is shown that small initializations ($\gamma > 0.5$) biases networks towards reasoning, i.e. the performance of tasks of type 1. accuracy increases more quickly as a function of training epochs than tasks of type 2. that compared to when a large initializations ($\gamma < 0.5$) is used.

This phenomenon is demonstrated empirically (transformers + a variation on set2vec, called Emb-MLP), as well as motivated theoretically using differential questions. The theoretical analysis highlights the distribution of task labels as a driving factor between the rates each task is solved.

## Update after rebuttal
Score increased to 4: Accept following resolutions to my questions during discussion phase.

**Claims And Evidence:**

Update: The author response has resolved claims that were partially supported.

The following claims are made, and are supported, or partially supported:
## Smaller initialization cases a "reasoning bias" in synthetic/controlled tasks (supported)

In the synthetic experiments, a smaller initialization ($\gamma=0.8$) shows reasoning tasks learning at a faster rate compared to the memorization experiments, whereas a larger initialization ($\gamma=0.3$) has the memorization tasks learning more quickly. This is shown both for transformers (Figure 2A) and Emb-MLP (2B), which is appears like a simple DeepSet variant [1].

[1] Deep Sets https://arxiv.org/abs/1703.06114

## The reasoning bias due to smaller initialization is a result of embedding space separation difference between the reasoning and memorization tasks in the chosen synthetic/controlled tasks (partially supported)

The authors show that reasoning-related tokens (anchors) become more distinct earlier in training than the memory tokens under smaller initialization ($\gamma=0.8$) (see Figure 3A). It takes a longer time at $\gamma=0.8$ for the memory tokens to become distinguishable. Since for the memory task, the memory tokens need to be distinguishable to be correctly retrieved, the embedding space structure learning directly corresponds to the ability to solve the task.

I note partially supported here, because we do not know what the rate of the embedding space learning looks like for other initialization scales. If e.g. Figure 3A looks similar for $\gamma=0.3$ I think that would reject this specific claim. This information is not provided in main text or in the appendix.

## The token label distribution affects the evolution of embeddings during training, and leads to the observed different rates of embedding structure discussed above (supported)

The authors present in Proposition 1 (+ Equation 5) the flow for the embedding vectors, and demonstrate the dominance by the label distribution. They derivate expressions for the label distributions in the memory task (Equation 7) and reasoning task (Equation 8). The empirical distribution for the reasoning task (Figure 3B, 3C) can be compared to the theoretical expression (Equation 8). They are consistent. The proofs is given in Appendix B.1 and B.2.

The proof of Lemma 2, and the distributions of the memory anchor seem correct. I was not able to verify the precise expanded form of the reasoning anchor distribution (Equations 22 - 25).

## The attention mechanisms allows a reasoning bias at small initialization scales (partially supported, unclear)

The paper demonstrates this through composition of two observations.

1. The paper demonstrates that the first attention mechanism behaves like an averaging operator. This is shown empirically in Figure 5, and with high probability in Lemma 1 (lines 380-384).
2. The $W_V$ projector has largest singular values aligned with the reasoning anchors, but nearly orthogonal to memory anchors. Consequently, the attention mechanism propagates reasoning anchor information, but not memory anchor information to subsequent tokens.

Point 1. is clear to me. The second I can see the evidence for in Figure 5C, and understand the projective behavior following the attention operation.

My challenge here is two-fold:
1. The transformer has a residual structure, which enables memory tokens to propagate to subsequent tokens (discussed at line 346, right hand column)
2. The "reasoning bias" discussed in this work relates to the rate at which a network learns certain information, and how this relates to the value of $\gamma$ used. In Figure 5 we are shown a single value of $\gamma$, and (I think) only the result at the end of training. We are not sensitive here to the rate at which phenomena occur, which is critical in the first two claims.

I can see that the large $\gamma$ value plays a role in the formal analysis, e.g. $\gamma\rightarrow\infty$ is an assumption in Propositions 2,3. It is unclear to me what happens away from this limit.

## The above observations translate to real world language tasks (partially supported)

The paper shows the embedding structure resulting from training a language model on a reasoning task (PrOntoQA) and memory task (TinyStories). The embedding structure presented is consistent with the synthetic analysis.

Only partially supported as the primary claim of this paper relates to the effect of initialization $\gamma$ on the reasoning bias of the model (measured as the rate of learning, which can be induced by the methods discussed in the above claims.)

The authors only show the results for $\gamma=0.8$ training, and do not show any analysis about how behavior changes as a function of training. To substantiate their claims, the authors need to present results from $\gamma=0.3$ and $\gamma=0.5$, and present any differences in the rates at which things happen for PrOntoQA and TinyStories that follow from the initialization scale change.

**Essential References Not Discussed:**

I am not aware of any essential missing references, although [3] may be useful to include for wider research framing.

[3] Understanding deep learning requires rethinking generalization https://arxiv.org/abs/1611.03530

**Experimental Designs Or Analyses:**

First, the analyses is in in general well done. There are certain aspects missing however, in general to draw a conclusion, in each case we need to see:
1. The $\gamma$ value varied
2. The rates of different phenomena change depending on the value of $\gamma$.
This type of analyses is only done for a subset of the experiments, and needs to be shown in all cases.

Second, the initialization scale is known to impact learning generally, see e.g. Maximal Update Parameterization ([2]). When we change the initialization scale, we expect to also need to change our learning rates in order to get a reasonable network behavior out. The experiments in the presented paper use the AdamW optimizer with a learning rate of 1.e-5 for every experiment, for every initialization. This is a suboptimal choice, and raises a small question regarding whether the phenomena observed in the paper would change if the learning rate was the optimal learning rate for each initialization.

[2] Tensor Programs V: Tuning Large Neural Networks via Zero-Shot Hyperparameter Transfer https://arxiv.org/abs/2203.03466

**Methods And Evaluation Criteria:**

The synthetic tasks constructed, as well as the chosen real-world tasks are sensible.

**Other Comments Or Suggestions:**

## Some suggestions for improving clarity

Try and bring Appendix A.1 into main text. If necessary, move some of the related work into the Appendix. Keep only related work elements critical to understanding the paper in main text, ideally directly compared to your own contributions.

Provide a roadmap for the reader.

In figures, state explicitly which $\gamma$ values are being used, and for which epochs. Do this for every figure.

Call out significant findings using \paragraph{...} notation, then back up the finding/observation with following prose.

Link to proofs from main text. E.g. in Proposition 1 (line 232) provide a ref to proof in the appendix. Do this for all results in main paper.

## Some suggestions for increasing confidence in claims

Present results for a range of $\gamma$ as a function of training for all experiments (see above discussion on claims).

Substantiate the LayerNormalization result that does not impact conclusion (line 375).

**Other Strengths And Weaknesses:**

The number of analyses provided from the different angles (empirical with many different measures, and theoretical) provides a rich, well-rounded view for the investigation.

Despite the many diagrams, I found the paper slightly challenging to read, potentially due to the sizeable introduction and related works sections. We only arrive at the contributions the others make in Section 3.1, which includes a highly technical presentation of the synthetic task (Appendix A.1 is much easier to follow), which took me significant work to understand the purpose of each aspect of the study.

**Questions For Authors:**

1. How critical is the transformer architecture in Section 4.3? Would a DeepSet [1] solve this task?
2. More generally one can initialize $W\in\mathbb R^{d_1\times d_2}, W_{i,j}\sim\mathcal N(\mu=0,\sigma=c \times d_2^{-\gamma})$ for some width-independent constant $c$ (see [2,4]). Why was the choice made to modify $\gamma$ as the controller of the initializer scale, rather than $c$ (optimal $\gamma$ choice is generally optimizer dependent, whereas $c$ is explicitly optimizer independent, so it would feel like varying $c$ is a more universal choice)? Is there an important difference between varying $\gamma$ or $c$? Is the choice stable across different model sizes?

[1] Deep Sets https://arxiv.org/abs/1703.06114

[2] Tensor Programs V: Tuning Large Neural Networks via Zero-Shot Hyperparameter Transfer https://arxiv.org/abs/2203.03466

[4] A Spectral Condition for Feature Learning https://arxiv.org/abs/2310.17813

**Relation To Broader Scientific Literature:**

This work is relevant in literature that investigates how networks learn, through either memorizing, or generalizing, e.g. [3], where networks can easily fit random labels when they are unstructured. The current paper shows the effect of initialization on the rate at which a model would memorize or generalize.

[3] Understanding deep learning requires rethinking generalization https://arxiv.org/abs/1611.03530

**Theoretical Claims:**

I validated:
- Proposition 1
- Equation 7
- Lemma 1

I was not able to validate the remaining theoretical claims due to their size and technical involvement within the reviewing period, however, I have no reason to expect they are incorrect.

I also note that while Lemma 1 is true at initialization, I am not sure if Lemma 1 is sufficient to guarantee that the attention mechanism remains an averaging operator throughout training (since in general, gradient updates disrupt initialization conditions). Figure 5A however empirically indicates that the attention mechanism is an averaging operator post-initialization.

---

> ### Author Rebuttal · Authors · 2025-03-31
>
> We sincerely appreciate your thorough review of our work and the valuable feedback you have provided. We address your concerns as follows: (**Due to the character limit, we are not able to display specific textual revisions. However, we will carefully revise our manuscript to address each concern.**)
>
> **P1: Different phenomena with different $\gamma$.**
>
> Given that our primary focus is to investigate the reasoning bias of models under small initialization scales, and due to the page limit, we present results for $\gamma=0.8$ in the main text. In Appendix C, we exhibit analogous analysis of the following components under $\gamma=0.3,0.5$:
>
> C.1: The embedding space of Emb-MLP,
>
> C.2: The embedding space of Transformer,
>
> C.3: The first attention module of Transformer.
>
> These analyses establish that varying initialization scales significantly influences the model’s behavior. We sincerely regret this omission of explicit references to Appendix C in the main text which will be provided in Sec.4.
>
> **P2: The real world language tasks.**
>
> Due to the same reason in P1, we did not include results for large initialization. However, we conducted experiments with $\gamma = 0.3, 0.5, 0.8$. We define the following metric: $\Delta L := \frac{L_{Tinystory}-L_{PrOntoQA}}{L_{PrOntoQA}}$. As $\gamma$ increases, $\Delta L$ exhibits an upward trend, indicating a growing bias for reasoning task (see https://postimg.cc/TpcZgmtw A). Analysis of embedding space during the early training stage (step 5000) aligns with that presented in Appendix. C (see B of the link above). These results will be added to the paper.
>
> **P3: The residual structure.**
>
> The residual structure operates through position-wise additive operations between two sequences, as:$residual(X,V)_i=X_i+V_i$，which lacks interactions between tokens at distinct positions.
>
> **P4: Attention module (figure 5) and Lemma.1.**
>
> We sincerely apologize for the omission of indicating epoch number in figure 5. Figure 5 displays the attention structure in the early training stage (epoch 200). By the end of training, the attention module exhibits specific patterns for capturing critical information within sequences. Lemma 1 only guarantees the average operator phenomenon during an early stage. Similar theoretical conclusions have been established in prior work, such as [1].
>
> **P5: The learning rate.**
>
> We conducted experiments with lr $\in[1e-5,5e-4]$. The learning bias under different $\gamma$ remains consistent across these configurations (see https://postimg.cc/7CtDSMV6 ). However, when lr increases to 1e-3, training becomes highly unstable, manifesting severe loss spike.
>
> **P6: Why assume $\gamma \to \infty$.**
>
> The assumption is primarily a technology adopted to enable asymptotic analysis in our theoretical framework. In finite-scale scenarios, we focus on the empirical trends as initialization scale decreases. Actually, for $\gamma\sim 1$, we can already see very clear reasoning bias in the training.
>
> **P7: LN does not impact conclusion.**
>
> We conducted an experiment with removing the LN module, exhibiting the same phenomena, i.e., smaller initialization scales bias reasoning task (see https://postimg.cc/dLbk0xyn ). Theoretically, we could provide an informal explanation. Since our analysis focuses on the initial stage of training, the mean and std could be approximated by the initial value. Consequently, the gradient flow is just multiplied with a constant 1/std, preserving the main structure of the learning dynamics. We will add these analyses in Appendix.
>
> **P8: Question 1.**
>
> In Sec.4.3, since noise and key tokens are sampled from the same distribution, the sequence lacks permutation invariance. To identify the key, the model must utilize positional encoding and components capable of cross-positional information exchange, like attention module. However, DeepSets is designed for permutation-invariant set input. It would fail to distinguish the key tokens. This fundamental limitation renders DeepSets unsuitable for this task.
>
> **P9: Question 2.**
>
> Prior work in [2,3] investigated the impact of $\gamma$ on model dynamics and identified distinct behavioral regimes, which is stable under different model sizes. Inspired by those works, we adopt to adjust $\gamma$.
>
> **P10: The suggestion on presentation.**
>
> We sincerely appreciate your suggestions and we will carefully consider and adopt these suggestions.
>
> **P11: Citation.**
>
> We appreciate your notification and will add this citation in Related Works.
>
> We once again extend our sincere gratitude for the valuable insights you have provided. We hope that our responses have addressed the concerns you raised.
>
>
> [1] Training Dynamics of Transformers to Recognize Word Co-occurrence via Gradient Flow Analysis. NeurIPS 2024.
>
> [2] Phase diagram for two-layer ReLU neural networks at infinite-width limit. JMLR 2021.
>
> [3] Initialization is Critical to Whether Transformers Fit Composite Functions by Reasoning or Memorizing. NeurIPS 2024.

---

> > ### Comment · Reviewer_FCaa · 2025-04-03
> >
> > Thank you for responding to my questions and comments, as well as those of the other reviewers. My review and score have been updated.

---

> > > ### Author Response · Authors · 2025-04-03
> > >
> > > Dear Reviewer FCaa,
> > >
> > > We sincerely appreciate your valuable suggestions and comments on our work. We are profoundly grateful for your recognition of our work and your willingness to update the score.
> > >
> > > The Authors.

---

### Official Review · Reviewer_a4CN · 2025-03-12

**Overall Recommendation:** 4

**Summary:**

This paper discusses the impact of initialization of language models on their trained performance on memorization and reasoning tasks. The paper uses proof to show reasoning tasks prefer smaller initialization while memorization tasks prefer large initialization. The authors attribute such behavior to being more differentiated in the embedding space at an early stage, which is further verified by empirical experiments.

**Claims And Evidence:**

The paper claims that reasoning tasks prefer smaller initialization while memorization tasks prefer large initialization, which is verified by theory and empirical experiments.

**Essential References Not Discussed:**

N/A

**Experimental Designs Or Analyses:**

The experimental design makes sense and supports the theoretical claims.

**Methods And Evaluation Criteria:**

This paper uses a synthetic dataset to verify their assumption. While data in natural language distribution (human annotation) will make the experiment more convincing, it's acceptable when such resource is absent.

**Other Comments Or Suggestions:**

N/A

**Other Strengths And Weaknesses:**

A potential weakness of this paper is not providing direct guidance for current LM fine-tuning as most tasks are now fine-tuned based on pre-trained models. The paper will be significantly more impactful if the discovery can be applied to analyzing the behavior of pre-trained LMs.

**Questions For Authors:**

N/A

**Relation To Broader Scientific Literature:**

This paper is related to the impact of model initialization on training performance.

**Theoretical Claims:**

I have checked the theoretical claims and think the proof to be convincing. But there is a chance that I miss minor mistakes.

---

> ### Author Rebuttal · Authors · 2025-03-31
>
> We sincerely appreciate your careful reading and evaluation of our work, and we are truly grateful for your recognition of our research. We address your concerns as follows.
>
> **Comment**: A potential weakness of this paper is not providing direct guidance for current LM fine-tuning as most tasks are now fine-tuned based on pre-trained models.
>
> **Response**: We appreciate this question you raised which is very interesting and valuable. We provide some preliminary intuitions: It is definitely important to investigate the impact of initialization on fine-tuning, and some related studies have already emerged in this field. For instance, [1] examined how parameter initialization in LoRA affects fine-tuning training. The study employed a small initialization scale $\gamma=1$ for matrix A, achieving more efficient feature learning. [2] theoretically explored the dynamic behavior of matrix factorization models under small initialization scales, demonstrating that small initialization reduces model complexity, then enhancing generalization capability. Due to the page limit in our current article, we leave this topic for future work. Actually, we have done some experiments on LoRA and believe our work can naturally be extended to the analysis of fine-tuning.
>
> **Comment**: While data in natural language distribution (human annotation) will make the experiment more convincing, it's acceptable when such resource is absent.
>
> **Response**: Thanks for your suggestion and we appreciate your understanding. In our experiments, we trained a standard GPT-2 model on two real-world datasets PrOntoQA and TinyStories though very limited resources. We observed that models with small initialization exhibited a learning bias toward reasoning tasks (see Figure 1). A parallel analysis of the embedding spaces for both datasets corroborated this conclusion (Sec.4.4).
>
> Thank you for your detailed feedback and for taking the time to review our work again. We are grateful if our responses have addressed your concerns.
>
> [1] The Impact of Initialization on LoRA Finetuning Dynamics. NeurIPS 2024.
>
> [2] Connectivity Shapes Implicit Regularization in Matrix Factorization Models for Matrix Completion. NeurIPS 2024.

---

> > ### Comment · Reviewer_a4CN · 2025-04-06
> >
> > Thanks for the further explanation. I believe the original version shall be accepted, and the follow-up modifications make the contribution clearer. Thank you!

---

> > > ### Author Response · Authors · 2025-04-07
> > >
> > > Dear Reviewer a4CN,
> > >
> > > We sincerely appreciate your thorough review and valuable feedback. Thank you for supporting our efforts, and we are pleased with your recognition of the modifications. Your insights have been instrumental in improving this work.
> > >
> > > The Authors.

---

### Decision · Program_Chairs · 2025-05-01

**Decision:**

Accept (spotlight poster)

**Comment:**

This work studies how the initialization scale affects the reasoning and memorization abilities of a model. For that, they use synthetic datasets, and show that small initializations promote reasoning behaviors. They also provide analysis on the separability of reasoning tokens compared to memorization tokens, showing that the former are more separable early on than the latter (under small init). The findings are complemented with theoretical claims that the reviewers (notably FCaa) found to be valid in general.


### Short summary of reviews

Reviewers a4CN and e1DG did not report many concerns. Reviewer a4CN commented on the use of synthetic data and reviewer e1DG on the generalization of the findings to RL methods. Both reviewers acknowledged a satisfactory rebuttal.

Reviewer FCaa provided an in-depth and very thorough review. They addressed the mathematical proofs and provided several suggestions. The authors gave a comprehensive rebuttal, leading to an score increase from 3 to 4.

### Recommendation

Given the positive scores (4, 4, 4) and the engagement in the rebuttal, I **recommend this paper for acceptance**.